# Cardiac disruption of SDHAF4-mediated mitochondrial complex II assembly promotes dilated cardiomyopathy

Xueqiang Wang[1,2,9], Xing Zhang[3,9], Ke Cao[2,9], Mengqi Zeng[2,9], Xuyang Fu[4,5,9], Adi Zheng[2], Feng Zhang[4,5], Feng Gao [4,5], Xuan Zou[6], Hao Li [2], Min Li[3], Weiqiang Lv[2], Jie Xu[2], Jiangang Long[2], Weijin Zang[7], Jinghai Chen [4,5], Feng Gao[3], Jian Ding [2✉], Jiankang Liu [1,2✉] & Zhihui Feng [1,8✉]

Succinate dehydrogenase, which is known as mitochondrial complex II, has proven to be a fascinating machinery, attracting renewed and increased interest in its involvement in human diseases. Herein, we find that succinate dehydrogenase assembly factor 4 (SDHAF4) is downregulated in cardiac muscle in response to pathological stresses and in diseased hearts from human patients. Cardiac loss of *Sdhaf4* suppresses complex II assembly and results in subunit degradation and complex II deficiency in fetal mice. These defects are exacerbated in young adults with globally impaired metabolic capacity and activation of dynamin-related protein 1, which induces excess mitochondrial fission and mitophagy, thereby causing progressive dilated cardiomyopathy and lethal heart failure in animals. Targeting mitochondria via supplementation with fumarate or inhibiting mitochondrial fission improves mitochondrial dynamics, partially restores cardiac function and prolongs the lifespan of mutant mice. Moreover, the addition of fumarate is found to dramatically improve cardiac function in myocardial infarction mice. These findings reveal a vital role for complex II assembly in the development of dilated cardiomyopathy and provide additional insights into therapeutic interventions for heart diseases.

[1] School of Health and Life Sciences, University of Health and Rehabilitation Sciences, 266071 Qingdao, Shandong, China. [2] Center for Mitochondrial Biology and Medicine, The Key Laboratory of Biomedical Information Engineering of Ministry of Education, School of Life Science and Technology, Xi'an Jiaotong University, 710049 Xi'an, Shaanxi, China. [3] Key Laboratory of Aerospace Medicine of the Ministry of Education, School of Aerospace Medicine, Fourth Military Medical University, 710032 Xi'an, China. [4] Department of Cardiology, Provincial Key Lab of Cardiovascular Research, Second Affiliated Hospital, Zhejiang University School of Medicine, 310009 Hangzhou, China. [5] Institute of Translational Medicine, Zhejiang University, School of Medicine, 310029 Hangzhou, China. [6] National & Local Joint Engineering Research Center of Biodiagnosis and Biotherapy, The Second Affiliated Hospital of Xi'an Jiaotong University, 710004 Xi'an, Shaanxi, China. [7] Department of Pharmacology, School of Basic Medical Sciences, Xi'an Jiaotong University Health Science Center, Xi'an, Shaanxi, China. [8] Frontier Institute of Science and Technology, Xi'an Jiaotong University, 710049 Xi'an, China. [9]These authors contributed equally: Xueqiang Wang, Xing Zhang, Ke Cao, Mengqi Zeng, Xuyang Fu. ✉email: jianding@xjtu.edu.cn; j.liu@mail.xjtu.edu.cn; zhfeng@mail.xjtu.edu.cn

Dilated cardiomyopathy (DCM) is characterized by left ventricle dilation and cardiac dysfunction, and can lead to substantial morbidity and mortality due to complications such as heart failure[1,2]. Primary DCM is considered idiopathic in the majority of cases, and the disease can also be caused by multiple cardiac insults, including myocardial ischemia and toxic, metabolic, and immunologic stresses. Extraordinary effort has been devoted to studying the mechanisms of DCM by identifying pathogenic genetic variants and nongenetic factors[3]. However, the molecular etiology and detailed mechanism remain relatively unclear[4].

Mitochondrial dysfunction and metabolic defects are associated with a series of cardiac diseases, including hypertrophic cardiomyopathy (HCM), DCM, arrhythmias, noncompaction of the ventricular myocardium, and heart failure[5–7]. Succinate dehydrogenase (SDH), which is the enzyme complex (referred to as mitochondrial respiratory complex II), represents a branch point and links the tricarboxylic acid cycle (TCA cycle, or Krebs cycle) with the electron transport chain, therefore occupying a very unique place in mitochondrial metabolism[8]. Recent studies have described the vital role of complex II signaling in modulating reactive oxygen species (ROS) production in various diseases and aging[9]. In the heart, decreased complex II activity was also observed in individuals with diabetic cardiomyopathy and hypertrophic cardiomyopathy.

A G555E mutation in the SDHA gene has been reported to cause DCM[10], thereby indicating that the normal function of complex II is required to maintain cardiac homeostasis. Intriguingly, in injured hearts and driven by accumulated succinate, the SDH complex may also mediate excess ROS production by reversing electron transport from complex II to complex I. The resulting oxidative damage drastically exacerbates cardiac remodeling[11,12]. Although the intricate involvement of complex II in the development of cardiomyopathy has been proposed in previous studies, the pathophysiological relevance remains poorly understood.

Complex II is comprised of four subunits: succinate dehydrogenase complex flavoprotein subunit A, B, C, and D (SDHA, SDHB, SDHC, and SDHD)[13]. Recent studies have revealed that the function of complex II relies on proper assembly of the SDH subunits mediated by the identified SDH assembly factors (SDHAFs)[14–16]. SDHAF4 is an assembly factor recently discovered in yeast[16]; however, the understanding of its pathological and physiological roles in mammals is largely insufficient. Herein, we report that SDHAF4 is substantially deregulated in response to cardiac stresses. Cardiac deletion of Sdhaf4 gene in mice suppressed complex II assembly and led to metabolic deficiency and activation of dynamin-related protein 1(Drp1), and induced excessive mitophagy in cardiac muscle, resulting in DCM and postnatal lethality. We also showed that targeting mitochondria through supplementation with fumarate, which was depleted due to insufficient succinate dehydrogenase activity, or Mdivi-1, an inhibitor of Drp1, substantially improved mitochondrial dynamics and cardiac function and prolonged the lifespan of DCM mice. This study reveals the vital role for mitochondrial complex II in cardiac pathogenesis and, more importantly, suggests potential therapeutic interventions for cardiac disorders.

## Results

### SDHAF4 is dysregulated in response to cardiac insult and in diseased hearts of human patients.
The heart is one of the organs with a high mitochondrial content, and cardiac muscle performance substantially relies on mitochondrial homeostasis. We speculate that SDHAFs may participate in regulating heart function. Four SDH assembly factors (SDHAF1-4) have been identified in mammals[17]. We assessed the expression of these SDHAFs in a myocardial infarction (MI) mouse model to obtain insights into their potential roles in cardiac remodeling and pathogenesis (Supplementary Fig. 1a). The expression of SDHAF 1, 2, 3, and 4 was monitored in mouse hearts (border regions) at 3, 14, and 28 days (MI_3d, MI_14d, and MI_28d), after MI surgery. The mRNA levels of Sdhaf1, 3 and 4 were altered in MI samples, while only the decrease in Sdhaf4 expression was statistically significant (Fig. 1a). We further examined the protein abundance of these SDH assembly factors using western blotting (Fig. 1b, c), immunohistochemistry (IHC, Supplementary Fig. 1b–e), and immunofluorescence (IF, Fig. 1d) staining. SDHAF4 was consistently downregulated in MI samples as demonstrated in all these assays. SDHAF4 was recently identified as a SDHAF in yeast[16], and immunofluorescence staining and microscopy confirmed the primary mitochondrial localization of SDHAF4 in mammalian cells (Fig. 1e). SDHAF4 was suggested to facilitate the assembly of SDHA with SDHB in yeast[16]. Notably, the interaction between SDHA and SDHB was decreased after MI surgery, suggesting a contributing role of downregulated SDHAF4 in MI mice (Fig. 1f, g). Meanwhile, we surveyed publicly available datasets of clinical samples (GSE135055)[18]. Remarkably, decreased expression of SDHAF4 in the heart appears to be closely associated with human cardiac disorders, including DCM (dilated cardiomyopathy) and MCD (microvascular coronary disease) (Fig. 1h). Together, these results indicate that SDHAF4 expression is very sensitive to cardiac insults. We speculate that SDHAF4 may be a unique factor of complex II that participates in regulating cardiac homeostasis. The study of SDHAF4 may provide an entry point for further assessments of the involvement of SDH in cardiac remodeling and pathogenesis.

### Cardiac loss of SDHAF4 results in heart defects and reduced postnatal viability.
We directly examined the role of SDHAF4 in the heart by producing a floxed allele of Sdhaf4 (Sdhaf4$^{fl}$) and subsequently generating mice in which the Ckmm-Cre transgene (expressing Cre recombinase driven by the muscle creatine kinase promoter) mediated the inactivation of Sdhaf4 in both cardiac and skeletal muscles (Supplementary Fig. 2a). The homozygous Sdhaf4 mutant mice (Sdhaf4$^{fl/fl,Ckmm}$-Cre; or Sdhaf4$^{Ckmm}$) had similar body weights to their wild-type (Sdhaf4-wild type, abbreviated hereafter as WT) littermates during the first 6 weeks after birth (Supplementary Fig. 2b). At an older age, Sdhaf4$^{Ckmm}$ mice started to show lower body weights than those of the controls. However, the observed reduced bodyweight at this age appears not to be the primary phenotype and is unlikely due to loss of Sdhaf4 in skeletal muscle per se, as the ratio of tibialis anterior (TA) muscle weight to bodyweight (BW) in mutants was comparable to that of WT littermates (Supplementary Fig. 2c).

Sdhaf4$^{Ckmm}$ exhibited significantly reduced viability, and none of the mutant mice survived beyond 12 weeks (Fig. 2a). Notably, the hearts of Sdhaf4$^{Ckmm}$ mice were substantially enlarged (Supplementary Fig. 2d), which suggests that these mutant mice have cardiac defects. We recorded surface electrocardiograms (ECGs) of these animals but did not detect significant abnormalities in the ECG parameters (Supplementary Fig. 2e). We also used echocardiography to measure cardiac function of Sdhaf4$^{Ckmm}$ and control mice. The heart rates of mutant animals were comparable to those of WT controls during the measurement (Supplementary Fig. 2f). However, a significant decrease in left ventricular fractional shortening (FS%) and deregulated cardiac contraction were detected in the mutants, indicating that the loss of Sdhaf4 impairs heart function (Fig. 2c–f).

Sdhaf4-floxed mice were crossed with tamoxifen (TAM)-inducible myosin heavy chain 6 (Myh6)-Cre transgenic mice (Mer-CreMer) to further verify the phenotypic consequences of SDHAF4 deficiency in the heart. The Cre recombinase mediated the deletion of Sdhaf4 specifically in cardiomyocytes of mice

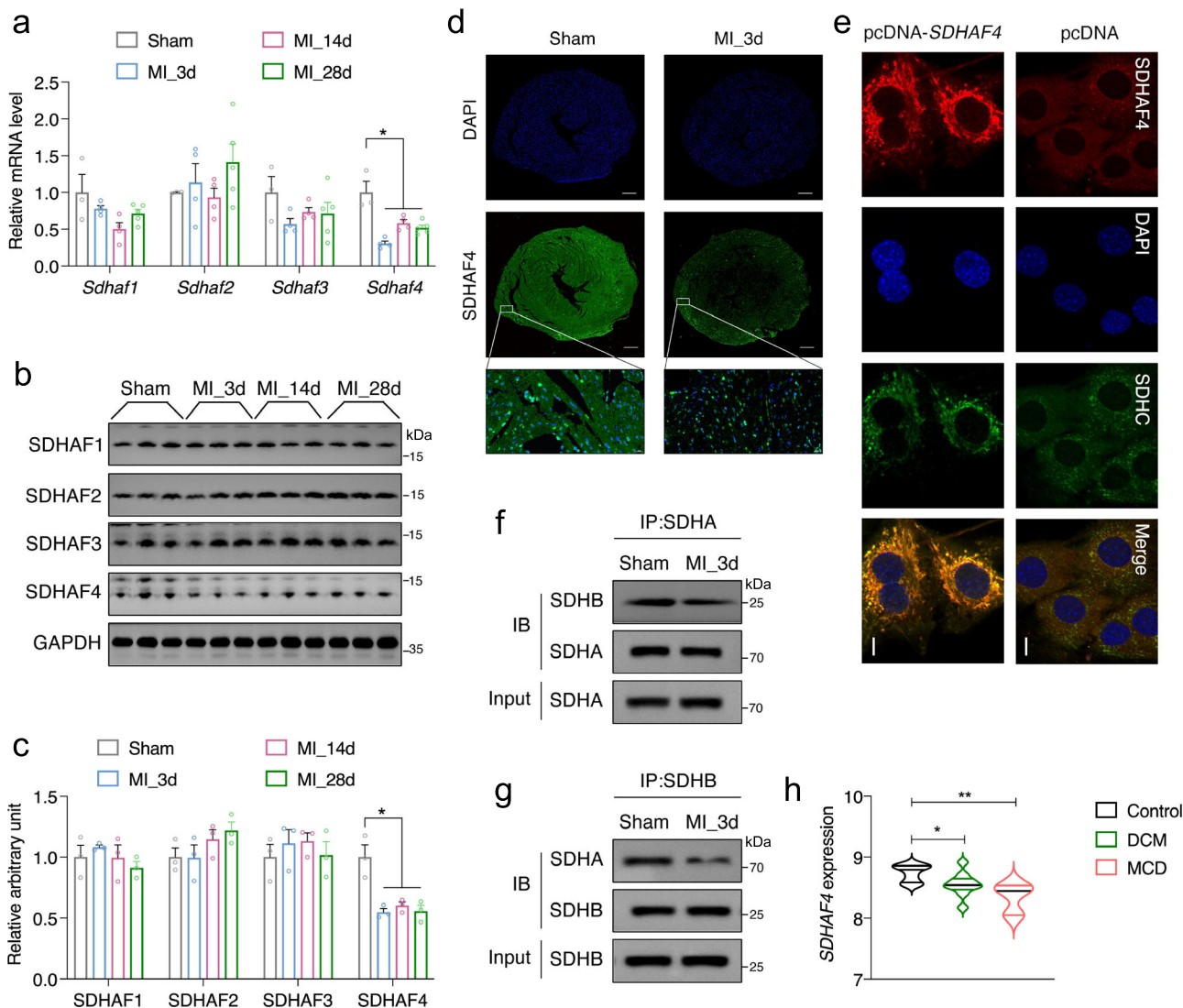

**Fig. 1 Downregulated SDHAF4 expression in myocardial infarction (MI) mice. a** Cardiac mRNA levels of *Sdhaf1*, *Sdhaf2*, *Sdhaf3*, and *Sdhaf4* in MI mice (Sham, $n = 3$; MI_3d, $n = 4$; MI_14d, $n = 4$; MI_28d, $n = 5$, $P = 0.013$). **b, c** Immunoblots for SDHAF1/2/3/4 expressions in left ventricles of MI mice at different stages (**b**, representative blotting images; **c** summary analysis of arbitrary unit), $n = 3$, $p = 0.015$. **d** Immunostaining of SDHAF4 in mouse cardiac section at day 3 after MI, scale bar, 2 mm ($n = 3$ biologically independent mice per group). **e** Immunostaining of SDHAF4 and SDHC for mitochondrial localization assessment, scale bar, 10 μm ($n = 3$ independent repeats per group). **f** Coimmunoprecipitation of SDHB with SDHA from the lysates of left ventricles in control and MI mice at day 3 after the surgery ($n = 3$ biologically independent mice per group). **g** Coimmunoprecipitation of SDHA with SDHB from the lysates of left ventricles in control and MI mice at day 3 after the surgery ($n = 3$ biologically independent mice per group). **h** Relative *SDHAF4* expression in the heart tissues of healthy, DCM and MCD patients from published database GSE135055, $n = 9$ for healthy control, $n = 10$ for DCM subjects, $n = 3$ for MCD subjects, $P = 0.028$, 0.0057. Imaging data was collected by ZEN 2012 Blue edition (ZEISS, Jena, Germany). Values are mean ± SEM, *$P < 0.05$, **$P < 0.01$. Statistical significance was determined one-way ANOVA with Tukey's multiple comparison test (**a, c**), and by two-tailed Student's *t*-test (**h**). Source data are provided in Source Data file. DCM dilated cardiomyopathy, MCD microvascular coronary disease.

supplemented with tamoxifen (Fig. 2g). After 8 weeks of tamoxifen induction, the homozygous mice (*Sdhaf4*fl/fl,Mer-CreMer, or *Sdhaf4*Mer-CreMer) presented significantly reduced viability. The mutant hearts were substantially enlarged (Fig. 2h, i). The phenotypes resembled what was observed in the *Sdhaf4*Ckmm mouse model (Supplementary Fig. 2d). Together, our results obtained from both *Sdhaf4*Ckmm and *Sdhaf4*Mer-CreMer animals consistently showed that abrogation of *Sdhaf4* led to cardiac abnormities and lethality, thereby indicating that *Sdhaf4* is an essential gene in the heart.

**Cardiac loss of *Sdhaf4* provokes dilated cardiomyopathy.** We next analyzed the gross morphology of the hearts at various

postnatal stages to monitor the pathogenic process in *Sdhaf4*Ckmm mice. *Sdhaf4*Ckmm mice had similar body weights to their WT littermates during the first 6 weeks after birth, while heart weight was substantially increased starting at the age of 3 weeks (Fig. 3a, b). Hematoxylin and eosin (H/E) histological staining confirmed these observations. *Sdhaf4*Ckmm animals started to display enlargement and thickening of cardiac muscle by 3–4 weeks, concurrent with the increased heart weight.

Dilation of the ventricular chambers was observed at later stages, suggesting that the loss of *Sdhaf4* in muscle results in progressive cardiac remodeling and DCM (Fig. 3c). Masson's trichrome staining showed no significant fibrosis in *Sdhaf4*Ckmm mice at an early age of 4 weeks, but evident collagen deposition was observed at a later age of 7 weeks, which was further

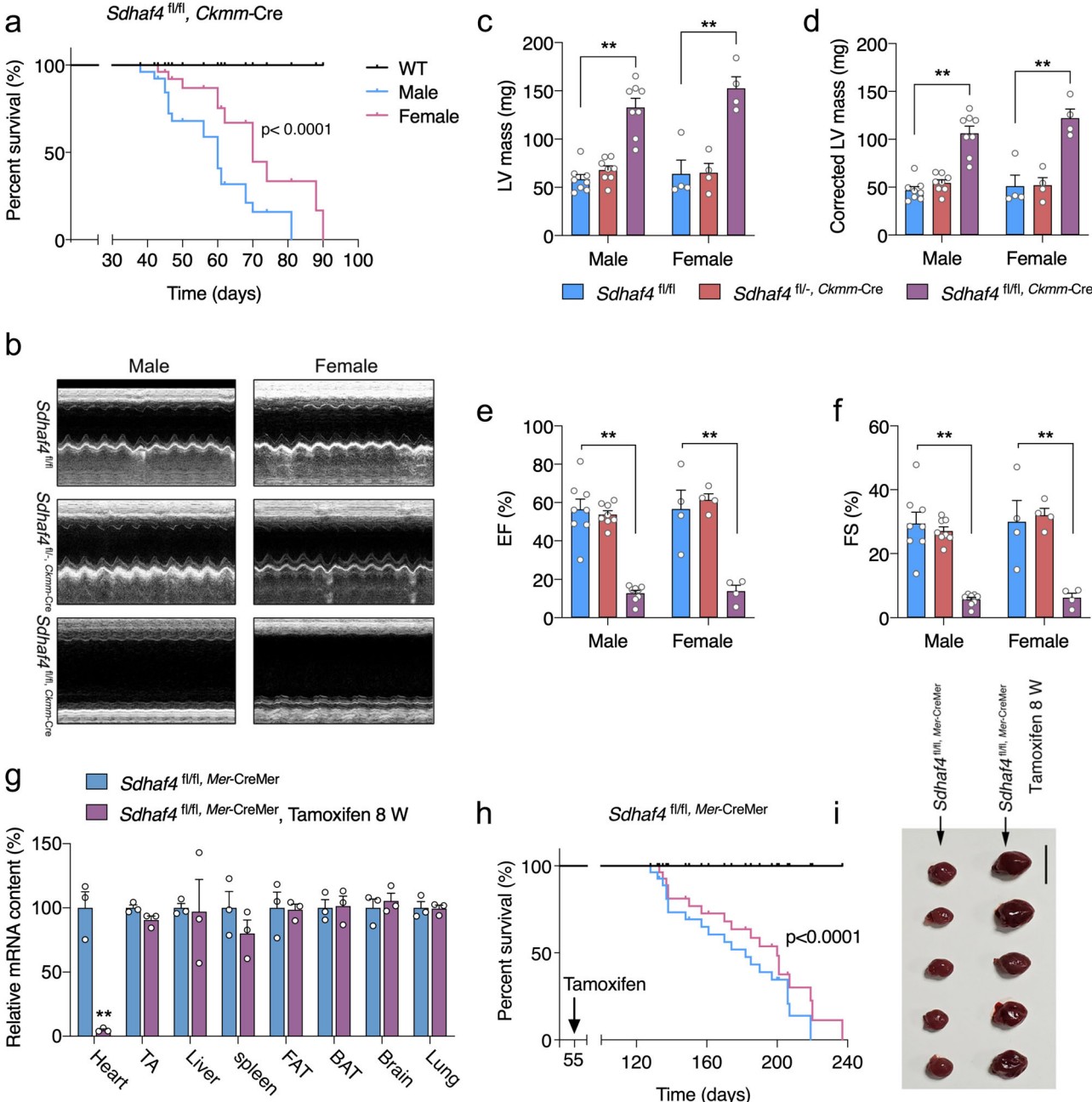

**Fig. 2 Cardiac loss of SDHAF4 shortens mouse lifespan due to cardiac failure. a** Postnatal survival curve for WT (*Sdhaf4*fl/fl, n = 15) and *Sdhaf4*fl/fl, Ckmm-Cre mice (male, n = 20; female, n = 11). **b–f** Heart function measurements of WT, heterozygous (*Sdhaf4*fl/-, Ckmm-Cre), and homozygous (*Sdhaf4*fl/fl, Ckmm-Cre) mutant mice: representative M-mode echocardiography (**b**), left ventricular mass (LV mass, **c**, P < 0.0001), corrected LV mass (**d**, P < 0.0001), percentage of ejection fraction (EF, **e**, P < 0.0001), percentage of fractional shortening (FS, **f**, P < 0.0001), n = 8 for male, n = 4 for female. **g** mRNA levels of *Sdhaf4* in tissues of WT (*Sdhaf4*fl/fl, Mer-CreMer, n = 3, P < 0.0001) and mutant mice (*Sdhaf4*fl/fl, Mer-CreMer after tamoxifen treatment for 8 weeks, n = 3). **h** Postnatal survival curve for WT (n = 15) and cardiac *Sdhaf4* mutant mice (male, n = 21; female, n = 18). **i** Heart images of WT and cardiac KO mice, scale bar, 1 cm. Values are mean ± SEM, *P < 0.05, **P < 0.01. Log-rank Mantel-Cox testing was performed for survival analysis (**a**, **h**), other statistical significance was determined by two-tailed Student's *t*-test (**c–g**). Source data are provided in Source Data file.

supported by Sirius red staining (Fig. 3d). Consistently, the expression of cardiomyopathy marker genes (fetal genes) was primarily increased in the left ventricles of *Sdhaf4*Ckmm mice at the age of 4 weeks (Fig. 3e). Upregulation of pathological markers was also observed in *Sdhaf4* Mer-CreMer mouse hearts (after 8 weeks of tamoxifen induction) (Fig. 3f). These results suggest that loss of *Sdhaf4* in the heart provokes cardiac pathogenesis. We further characterized the underlying molecular events by performing RNA-seq to profile the transcriptomes of *Sdhaf4*Ckmm

and control hearts (6 weeks old) and identified 1117 upregulated and 1035 downregulated genes involved in multiple aspects of biological processes (Supplementary Fig. 3a–d). The KEGG (Kyoto Encyclopedia of Genes and Genomes) pathway enrichment analysis showed that the HIF-1 (hypoxia-induced factor 1) signaling pathway was upregulated (Supplementary Fig. 3e, f). HIF signaling is induced by cardiac dysfunction, and its activation has been observed in various cardiomyopathies, including ischemic heart disease, DCM and heart failure[19]. Previous studies

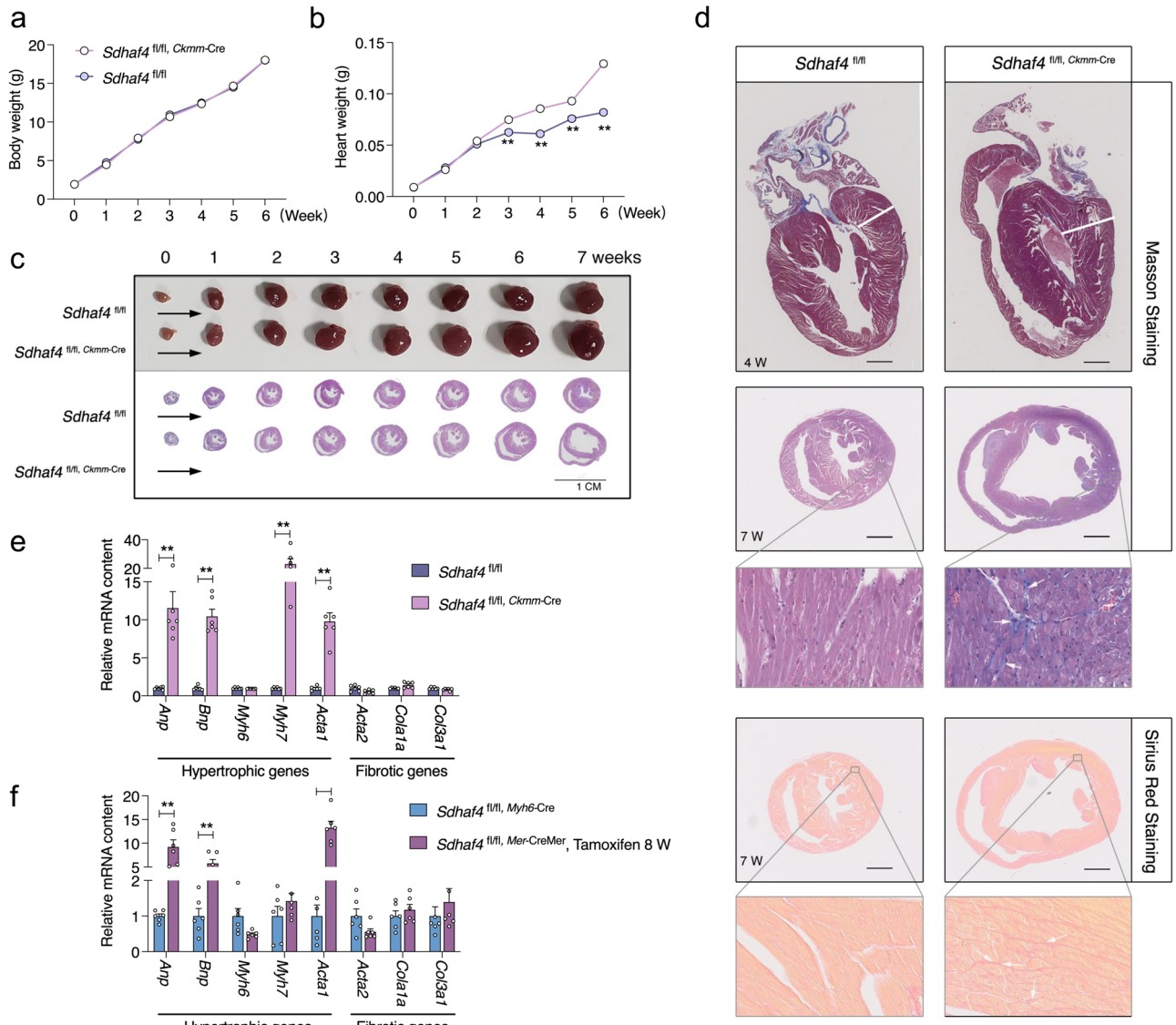

**Fig. 3 Cardiac loss of SDHAF4 provokes dilated cardiomyopathy. a** Postnatal bodyweight curves of WT (*Sdhaf4*fl/fl, $n = 6$) and *Sdhaf4* mutant mice (*Sdhaf4*fl/fl, Ckmm-Cre, $n = 6$). **b** Postnatal heart weight curves of WT ($n = 6$) and *Sdhaf4* mutant mice ($n = 6$), $P = 0.002, 0.0001, 0.0018, 0.0001$. **c** Macroscopic pictures and H/E histology of hearts of WT and mutant mice, scale bar, 1 cm. **d** Representative cardiac Masson trichrome staining of WT and mutant mice at age of 4 and 7 weeks, and Sirius red staining of WT and mutant mice at age of 7 weeks, scale bar, 3 mm. **e** Relative expressions of hypertrophic and fibrotic-associated genes in left ventricle of WT and muscle *Sdhaf4* mutant mice, $n = 6$, $P < 0.0001$. **f** Relative expressions of hypertrophic and fibrotic-associated genes in left ventricle of WT and cardiac *Sdhaf4* mutant mice, $n = 6$, $P < 0.0001$. Values are mean ± SEM, *$P < 0.05$, **$P < 0.01$. Statistical significance was determined by two-tailed Student's *t*-test (**b**, **e**, **f**). Source data are provided in Source Data file.

have also documented the cardioprotective effect of the HIF pathway on various cardiac disorders[20]. The induction of HIF signaling, therefore, may represent an adaptive response in the hearts of mutant animals.

As displayed in the KEGG pathway enrichment analysis, metabolic pathways, TCA cycle and oxidative phosphorylation (OxPhos) were significantly downregulated in *Sdhaf4* mutant mouse hearts. Deregulation of the representative genes in these pathways was further validated using qPCR (Supplementary Fig. 4a). Next, we compared the differentially expressed gene (DEG) list in the *Sdhaf4*Ckmm hearts with previously reported gene profiling datasets of MI[21]. Notably, the expression of 31.07% (749/2420) of the upregulated genes in MI (MI_3d) was increased in *Sdhaf4* mutants (Supplementary Fig. 4b). Among the downregulated genes in the MI model, 39.53% (1065/2694) also displayed reduced expression in *Sdhaf4*Ckmm hearts

(Supplementary Fig. 4b). In particular, downregulation of TCA cycle-related genes was consistently observed in both datasets. We also resurveyed these TCA genes in a profiling data of DCM models[22]. Intriguingly, these genes were consistently downregulated in DCM hearts (Supplementary Table 1). The results suggest that similar or common molecular processes occur in MI, DCM, and *Sdhaf4*Ckmm hearts, further indicating that metabolic deregulation in the mutant hearts may be one of the key processes relevant to cardiac pathogenesis.

**Disrupted complex II assembly leads to mitochondrial dysfunction in the heart.** SDHAF4 was shown to interact specifically with the catalytic SDHA subunit and facilitate its association with SDHB for the subsequent assembly of the SDH complex[16]. We measured the expression of SDH components in mouse hearts at

various stages. The relative abundance of SDHAF4 and other SDH complex subunits gradually increased during postnatal development (Fig. 4a). Loss of *Sdhaf4* in the heart (*Sdhaf4$^{Ckmm}$*) resulted in specifically decreased levels of SDH complex subunits at an early postnatal age of 3 weeks (Fig. 4b, c and Supplementary Fig. 5a). The abundance of complex I, III, IV, and V components was not altered in the mutant hearts at this age (Fig. 4b, c and Supplementary Fig. 5a). Despite the reduced protein levels, the mRNA abundance of SDH subunits remained unchanged in the *Sdhaf4$^{Ckmm}$* hearts. We surmise that the deregulation of complex II is very likely to occur at the posttranslational level and is primarily due to assembly defects (Fig. 4d).

Next, we assessed the effects of SDHAF4 deficiency on complex assembly. Immunoprecipitation assays using cardiac muscle tissues showed that abrogation of *Sdhaf4* attenuated the interaction between SDHA and SDHB (Fig. 4e, f), which may lead to SDH subunit degradation, as evidenced by increased levels of the ubiquitin modification (Fig. 4g). In addition, the formation of mitochondrial supercomplexes was hindered, and defects in mitochondrial oxygen consumption were observed in the mutant hearts (Fig. 4h–j). Intriguingly, the relative ATP level remained unchanged 3 weeks after birth, although it was slightly decreased in the hearts of 8-week-old *Sdhaf4$^{Ckmm}$* mice (Fig. 4k). Notably, the levels of subunits of other mitochondrial complexes were decreased later at the age of 8 weeks, accompanied by a global reduction in complex activities, which may be secondarily due to the turnover of complex II (Supplementary Fig. 5c–e). Meanwhile, a substantial increase in protein oxidation was detected in the mitochondrial fraction of cardiac lysates at the early and postnatal age of 3 weeks and was also present at the age of 8 weeks (Supplementary Fig. 5f, g), thereby indicating that mitochondrial oxidative stress occurred in the mutant mice.

Since complex II is the key factor that couples mitochondrial electron transport and the TCA cycle[23], we asked whether inactivation of SDHAF4 deficiency altered the abundance of key enzymes and metabolites of the TCA cycle. The protein levels of key TCA cycle-related enzymes in the mutant hearts at an early age (3 weeks after birth) appeared to be indistinguishable from those in control animals, whereas at a later age of 8 weeks, the hearts exhibited a global decrease in the levels of these enzymes (Fig. 5a, b). Additionally, the levels of TCA cycle-related metabolites, including pyruvate, citrate, isocitrate, malate, and fumarate, were substantially decreased, except that succinate was upregulated in the mutant mice (Fig. 5c). These alterations appear to be the primary effects resulting from the loss of SDH activity, which is required for catalyzing the oxidation of succinate into fumarate, a key step in the TCA cycle. Further metabolomics analysis revealed that multiple amino acid metabolic pathways were affected by the SDHAF4 deficiency, among which the arginine biosynthesis pathway was the most significantly altered pathway (Fig. 5d, e and Supplementary Fig. 6a–d). A link between the TCA cycle and arginine biosynthesis has been reported recently. Arginine is regenerated through the citrulline-nitrogen monoxide (NO) pathway, in which citrulline and aspartate are converted to arginine by argininosuccinate synthase (ASS) and argininosuccinate lyase (ASL) with fumarate as a byproduct, while malate, the product of fumarase action on fumarate, is converted to oxaloacetate for the production of aspartate to join the arginine biosynthesis cycle[24] (Fig. 5f). Thus, the SDHAF4 deficiency not only primarily altered the levels of TCA cycle-related metabolites but also suppressed arginine regeneration, which further exacerbated metabolic stress.

## Disrupted complex II assembly alters mitochondrial fusion/fission dynamics in the heart.

Despite impaired mitochondrial function, the morphology of the organelle was not altered in

*Sdhaf4$^{Ckmm}$* mice at the age of 3 weeks (Fig. 6a). As shown in the transmission electron microscopy (TEM) images, the mitochondrial number and their subcellular distributions were comparable in both WT and *Sdhaf4$^{Ckmm}$* mice (Fig. 6b–e). Notably, during pathogenesis, the cardiomyocytes of *Sdhaf4$^{Ckmm}$* mice at the age of 8 weeks presented severe morphological abnormalities in myofibril/sarcomere organization and intramitochondrial structure, including cristae malformations, electron-dense inclusions and hypodense compartments (Fig. 6f, g). Interestingly, the total number of mitochondria was significantly increased in *Sdhaf4$^{Ckmm}$* mice, while the size was substantially reduced (Fig. 6h–j). We asked whether the increased number but reduced size of mitochondria in the *Sdhaf4$^{Ckmm}$* cardiomyocytes was potentially due to alterations in mitochondrial biogenesis or fusion/fission processes. However, the qPCR analysis revealed a decrease in the relative abundance of mtDNA instead of an increase in mutant cardiac muscle (Fig. 6k). We also measured the levels of PGC-1α and several other regulators of mitochondrial biogenesis and found that their expression was reduced in *Sdhaf4$^{Ckmm}$* mice (Fig. 6l, m). Therefore, the increased quantity of these organelles is not likely to be resulted from changes in mitochondrial biogenesis. Next, the protein levels of numerous factors regulating mitochondrial fusion and fission were assayed, yet no significant differences were detected between control and mutant hearts. However, the posttranslational modifications of Drp1, which is a key factor regulating mitochondrial fission, were altered in the *Sdhaf4$^{Ckmm}$* mice. The activity of Drp1 strongly depends on the phosphorylation status of two serine sites. Phosphorylation at serine616 (p-Drp1$^{s616}$) enhances mitochondrial fission[25], while serine637phosphorylation (p-Drp1$^{s616}$) represses mitochondrial fission activity[26]. Increased phosphorylation of p-Drp1$^{s616}$ but decreased p-Drp1$^{s637}$ levels were observed in the hearts of 3-week-old *Sdhaf4*-KO mice (Fig. 6n). Notably, Drp1 activation was also observed in adult *Sdhaf4$^{Ckmm}$* mice (Fig. 6n, p), suggesting an early and sustained activation of Drp1 in the absence of SDHAF4. Together, these results indicate that the SDHAF4 deficiency may enhance mitochondrial fission, potentially resulting in the fragmentation of the organelles. Mitophagy usually couples with mitochondrial fusion/fission dynamics to enable the recycling and clearance of damaged organelles. Intriguingly, the level of LC3-II (microtubule-associated protein 1 light chain 3 conversion product), a hallmark of autophagy activation[27,28], was substantially increased in adult *Sdhaf4$^{Ckmm}$* hearts (Fig. 6o, p). We further confirmed this observation by assessing mitophagy in the H9C2 rat cardiomyocyte cell line using a fluorescent reporter (mt-Keima)-based approach. Keima is a lysosomal protease-resistant, dual-excitation ratiometric fluorescent protein that is sensitive to changes in pH. Fusion with a mitochondrial signal peptide enables the protein (mt-Keima) to be localized to mitochondria, where the pH value is normally 8.0, and the short wavelength excitation of Keima predominates. Once mitochondria are taken up by acidic lysosomes (pH 4.5) during mitophagy, mt-Keima will undergo a gradual shift to longer wavelength excitation. The mt-Keima reporter therefore allows us to more readily monitor mitophagy in cells[29]. Indeed, knockdown of *Sdhaf4* substantially increased mitophagosome formation (Fig. 6q), further confirming that SDHAF4 deficiency accelerated mitophagy in cardiomyocytes.

Drp1 activation was observed in the mutant cardiac muscle prior to the global downregulation of mitochondrial complexes and the pathological alteration in gross heart morphology, thus appeared to be one of the initial molecular events that may trigger the subsequent cardiac pathogenesis. We asked how the Drp1 phosphorylation was regulated in the hearts of *Sdhaf4$^{Ckmm}$* mice. Adenosine monophosphate-activated protein kinase, AMPK, has been reported to sense a decrease in cellular energy and activate

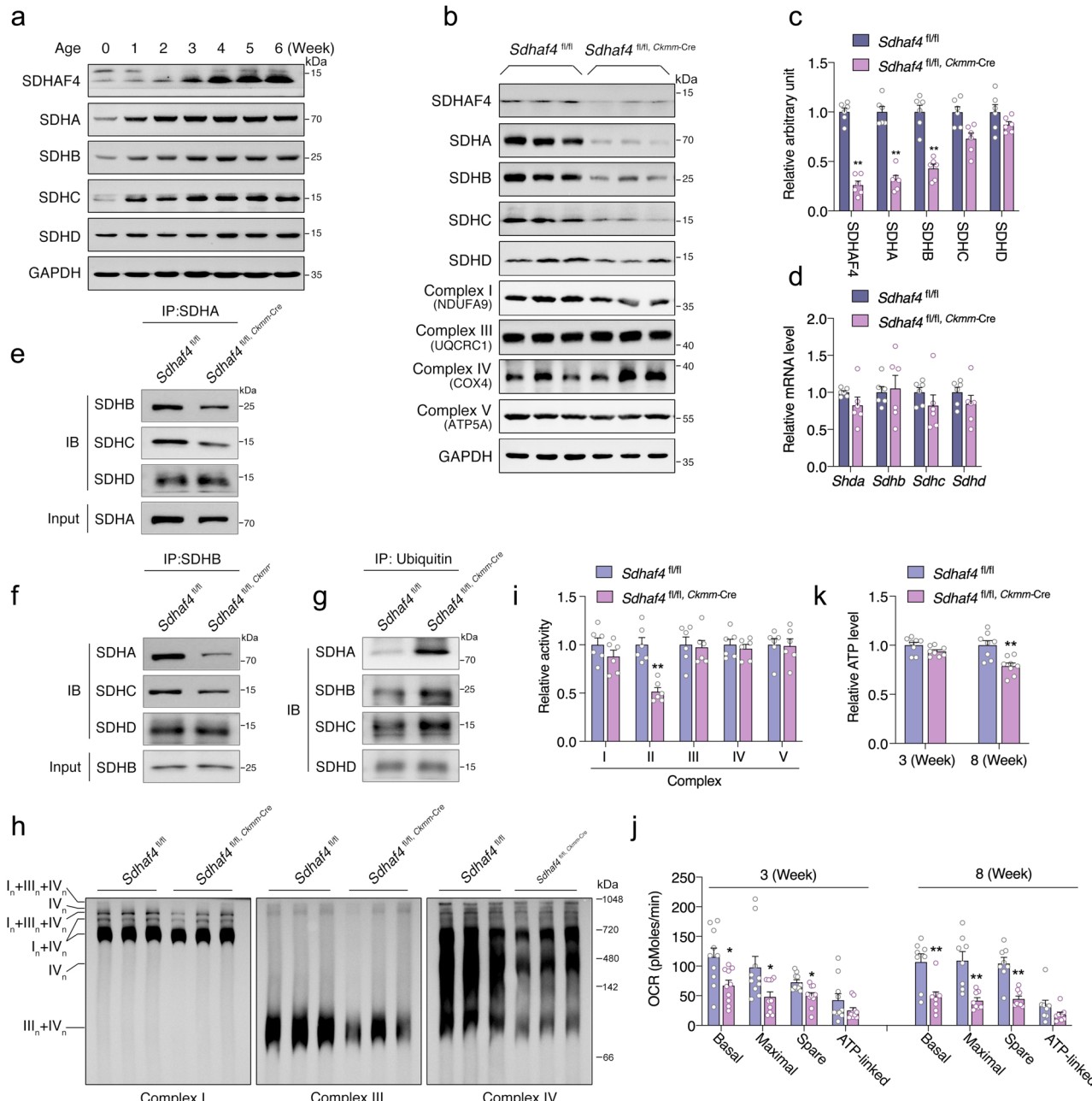

**Fig. 4 Loss of SDHAF4 disrupts complex II assembly and promotes mitochondrial dysfunction. a** Immunoblots for SDHAF4 and complex II subunits in left ventricles of WT mice at different ages. **b, c** Immunoblots for SDHAF4 and mitochondrial complex subunits in left ventricles of hearts from WT and *Sdhaf4* mutant mice at age of 3 weeks (**b**, representative blotting image; **c**, summary analysis of arbitrary unit, n = 6, P < 0.0001). **d** mRNA levels of *Sdha, Sdhb, Sdhc,* and *Sdhd* in left ventricles of hearts from WT and *Sdhaf4* mutant mice (n = 6). **e** Coimmunoprecipitation of SDHB, SDHC, and SDHD with SDHA from the lysates of left ventricles of hearts from WT and *Sdhaf4* mutant mice at age of 3 weeks. **f** Coimmunoprecipitation of SDHA, SDHC, and SDHD with SDHB from the lysates of left ventricles of hearts from WT and *Sdhaf4* mutant mice at age of 3 weeks. **g** Coimmunoprecipitation of SDHA, SDHB, SDHC, and SDHD with Ubiquitin from the lysates of left ventricles of hearts from WT and *Sdhaf4* mutant mice at age of 3 weeks. **h** Immunoblots for mitochondrial supercomplex: mitochondrial proteins of left ventricles of hearts from WT and *Sdhaf4* mutant mice were subjected to BN-PAGE and immunoblot with NDUFA9 (subunit of Complex I), UQCRC1 (subunit of Complex III), and COX4 (subunit of complex IV) antibodies. **i**, Analysis of complex activities of mitochondrion from left ventricles of hearts from WT and *Sdhaf4* mutant mice at age of 3 weeks, n = 6, P = 0.0002. **j-k** Mitochondrial oxygen consumption rate (OCR, P = 0.013, 0.026, 0.015, 0.003, 0.001, 0.00013) and ATP level of left ventricles of hearts (P = 0.002) from WT and *Sdhaf4* mutant mice at age of 3 and 8 weeks, n = 9. Values are mean ± SEM, *P < 0.05, **P < 0.01. Statistical significance was determined by two-tailed Student's t-test (**c, d, i-k**). Source data are provided in Source Data file.

Drp1. However, neither the total AMPK abundance nor phosphorylated AMPK levels, in cardiac muscle samples harvested from 3-week-old *Sdhaf4*[Ckmm] mice, were altered (Fig. 6l). This finding is consistent with the results described above, in which no alteration in ATP levels was detected in the

hearts of *Sdhaf4*[Ckmm] mice at the same age (Fig. 4k). Thus, the observed phosphorylation of Drp1 did not appear to depend on AMPK activity.

Next, we aimed to test the effect of extracellular signal-regulated protein kinase, ERK1/2, which has been suggested to promote Drp1-

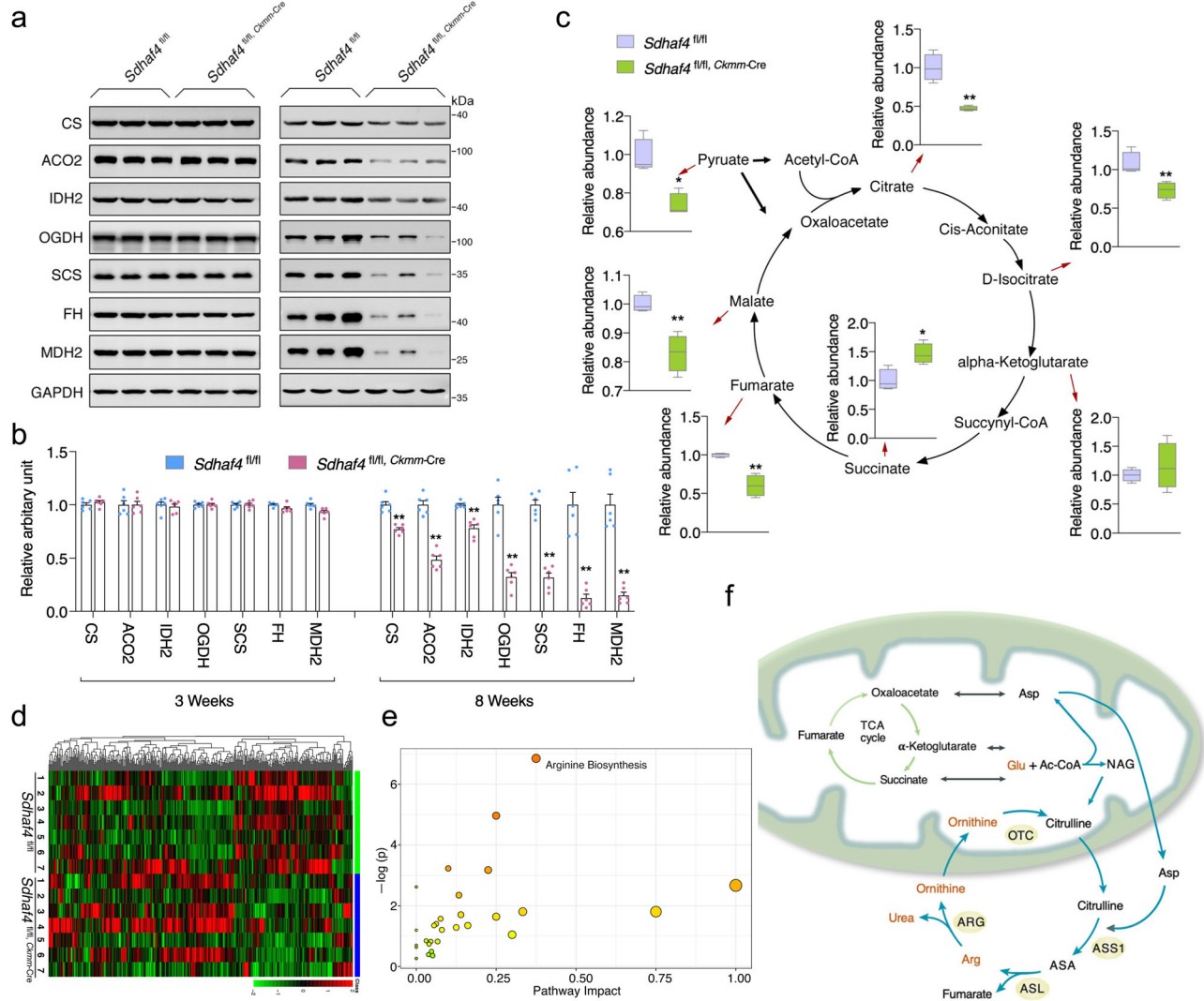

**Fig. 5 Loss of SDHAF4 exacerbates TCA cycle-associated metabolic capacity. a, b** Immunoblots for TCA cycle enzymes in left ventricles of hearts from WT and *Sdhaf4* mutant mice at age of 3 and 8 weeks respectively, representative blotting images (**a**), summary analysis of arbitrary unit (**b**), *n* = 6, *P* < 0.0001. **c** Quantitative measurements of TCA cycle metabolites pyruvate, malate, fumarate, succinate, citrate, isocitrate, and alpha-ketoglutarate (*n* = 4, *P* = 0.012, 0.001, 0.0013, 0.009, 0.003). Box plots indicate median (middle line), 25th, 75th percentile (box), and 5th and 95th percentile (whiskers). **d–f** Metabolomics analysis for left ventricles of WT and *Sdhaf4* mutant mice at age of 8 weeks, with *Z*-score plot of global metabolic profiles among all the samples (**d**), metabolic pathway enrichment analysis (**e**), and representative images of alterations on arginine biosynthesis pathway (**f**, green arrows for TCA cycle; blue arrows for arginine biosynthesis pathway; red highlights decreased metabolites), *n* = 7. Values are mean ± SEM, *\*P* < 0.05, *\*\*P* < 0.01. Statistical significance was determined by two-tailed Student's *t*-test (**b**, **c**). Source data are provided in Source Data file. TCA cycle tricarboxylic acid cycle.

dependent mitochondrial fission[30]. Knockdown of *Sdhaf4* in H9C2 cells substantially increased ERK1/2 phosphorylation, accompanied by increased p-Drp1[s616] and LC3-II levels, which were all reversed by U0126, a specific inhibitor of ERK1/2 (Supplementary Fig. 7a, b). Consistent with these results, increased p-ERK1/2 levels were observed in *Sdhaf4*[Ckmm] mice (Supplementary Fig. 7c, d). Administration of U0126 to *Sdhaf4*[Ckmm] mice completely blocked Drp1 phosphorylation and attenuated mitophagy in the heart (Supplementary Fig. 7c, d). Thus, ERK1/2 can mediate Drp1 activation and excess mitophagy in *Sdhaf4*[Ckmm] mice.

**Mitochondria-targeted interventions prolong the lifespan of mice with DCM.** Based on the aforementioned observations, we speculated that an impaired metabolic capacity and accelerated fission/mitophagy may be the two major mitochondrial events underlying the onset of DCM in *Sdhaf4*[Ckmm] animals; thus, a

targeted metabolic or chemical intervention was assumed to be beneficial. We tested this hypothesis by treating the mice with sodium fumarate dibasic through oral water intake or supplemented the animals with a specific Drp1 inhibitor, Mdivi-1, through intraperitoneal injection. Notably, as revealed by the TEM analysis, mitochondrial and myofiber integrity were improved by supplementation with either fumarate or Mdivi-1 (Fig. 7a). The mitochondrial number per section and the average size were substantially restored as well (Fig. 7b, c). The *Anp* (natriuretic peptide type A) mRNA level was decreased in the two intervention groups, and the upregulation of *Bnp* (natriuretic peptide type B) and *Myh7* (myosin heavy chain 7) was attenuated by the Mdivi-1 treatment (Fig. 7d). Echocardiography measurements showed that supplementation with Mdivi-1 significantly improved the heart function of *Sdhaf4*[Ckmm] mice. A slight increase in the factional shortening (FS%) value was observed in

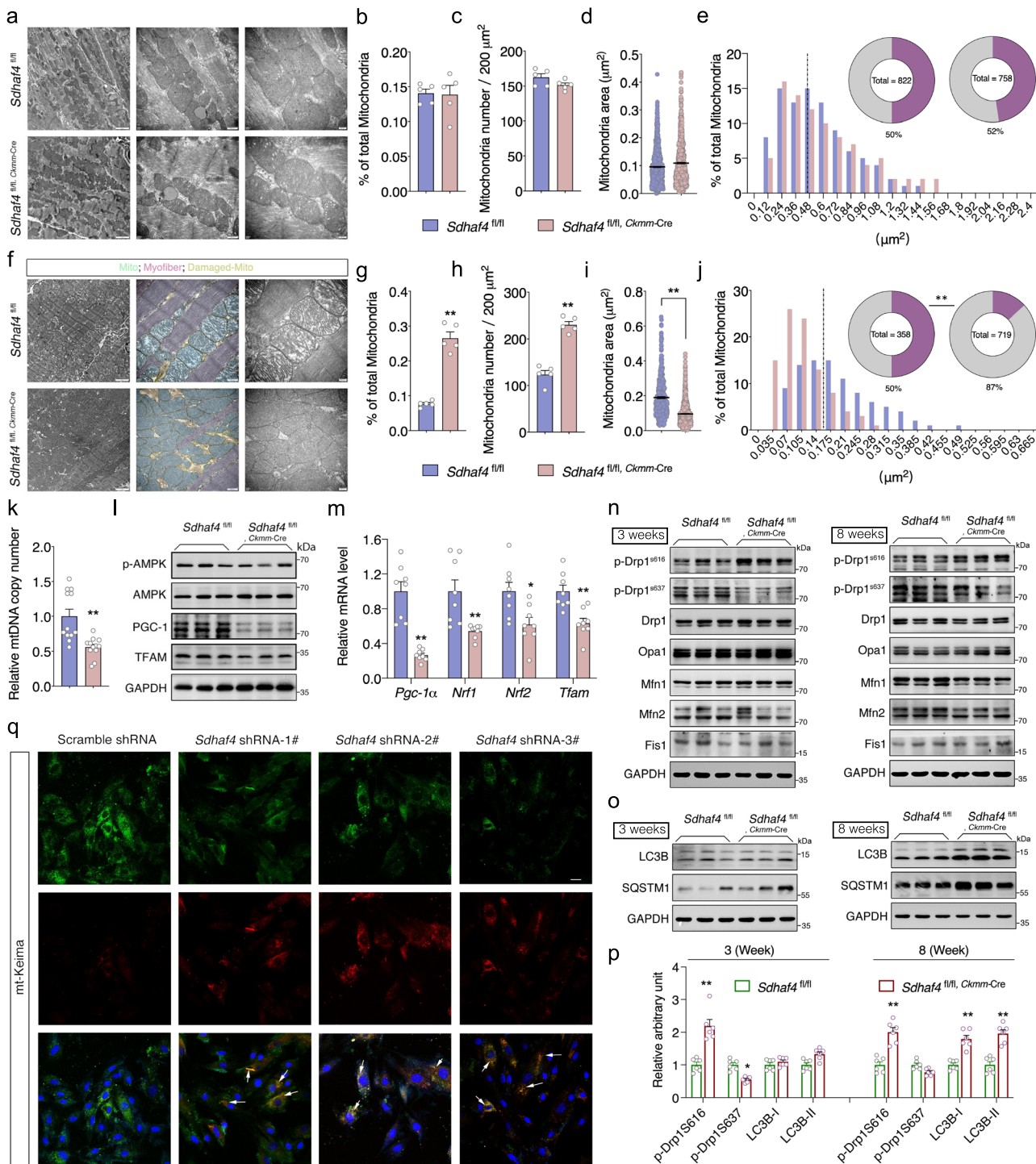

**Fig. 6 Disrupted complex II assembly promotes mitophagy in heart. a–e** Transmission electron microscope (TEM) analysis for left ventricles of hearts from WT and *Sdhaf4* mutant mice at age of 3 weeks, representative images (**a**, scale bar, 1 μm, 500 nm, and 200 nm, respectively), ratio of abnormal mitochondria (**b**, $n = 5$), mitochondrial number per section (**c**, $n = 5$), average mitochondrial area (**d**, $n = 822$ for WT, $n = 758$ for mutants), distribution pattern of mitochondrial area, (**e**, $n = 822$ for WT, n = 758 for mutants). **f–j** TEM analysis for left ventricles of hearts from WT and *Sdhaf4* mutant mice at age of 8 weeks, representative images (**f**), ratio of abnormal mitochondria (**g**, $n = 5$, $P < 0.0001$), mitochondrial number per section (**h**, $n = 5$, $P < 0.0001$), average mitochondrial area (**i**, $n = 358$ for WT, $n = 719$ for mutants, $P < 0.0001$), distribution pattern of mitochondrial area, (**j**, $n = 358$ for WT, $n = 719$ for mutants). **k–m** mtDNA copy number $n = 12$ ($P = 0.0003$), immunoblots ($n = 3$) and qPCR analysis ($n = 8$, $P = 0.00001$, 0.005, 0.012, 0.001) of mitochondrial biogenesis related genes. **n–p** Immunoblots for mitochondrial fusion and fission related proteins, LC3B and SQSRM1 in left ventricles of hearts from WT and *Sdhaf4* mutant mice at age of 3 and 8 weeks respectively (**n**, **o** representative blotting image; **p** summary analysis of arbitrary unit, $n = 6$, $P = 0.0002$, 0.011, 0.0001, 0.00011, 0.00001). **q** mt-Keima imaging in H9C2 cells transfected with *Sdhaf4* shRNAs, arrows indicate mitophagy puncta, imaging data was collected by ZEN 2012 Blue edition (ZEISS, Jena, Germany), scale bar, 20 μm ($n = 3$ independent repeats per group). Values are mean ± SEM, *$P < 0.05$, **$P < 0.01$. Statistical significance was determined by two-tailed Student's *t*-test (**g–i**, **k**, **m**, **p**). Source data are provided in Source Data file.

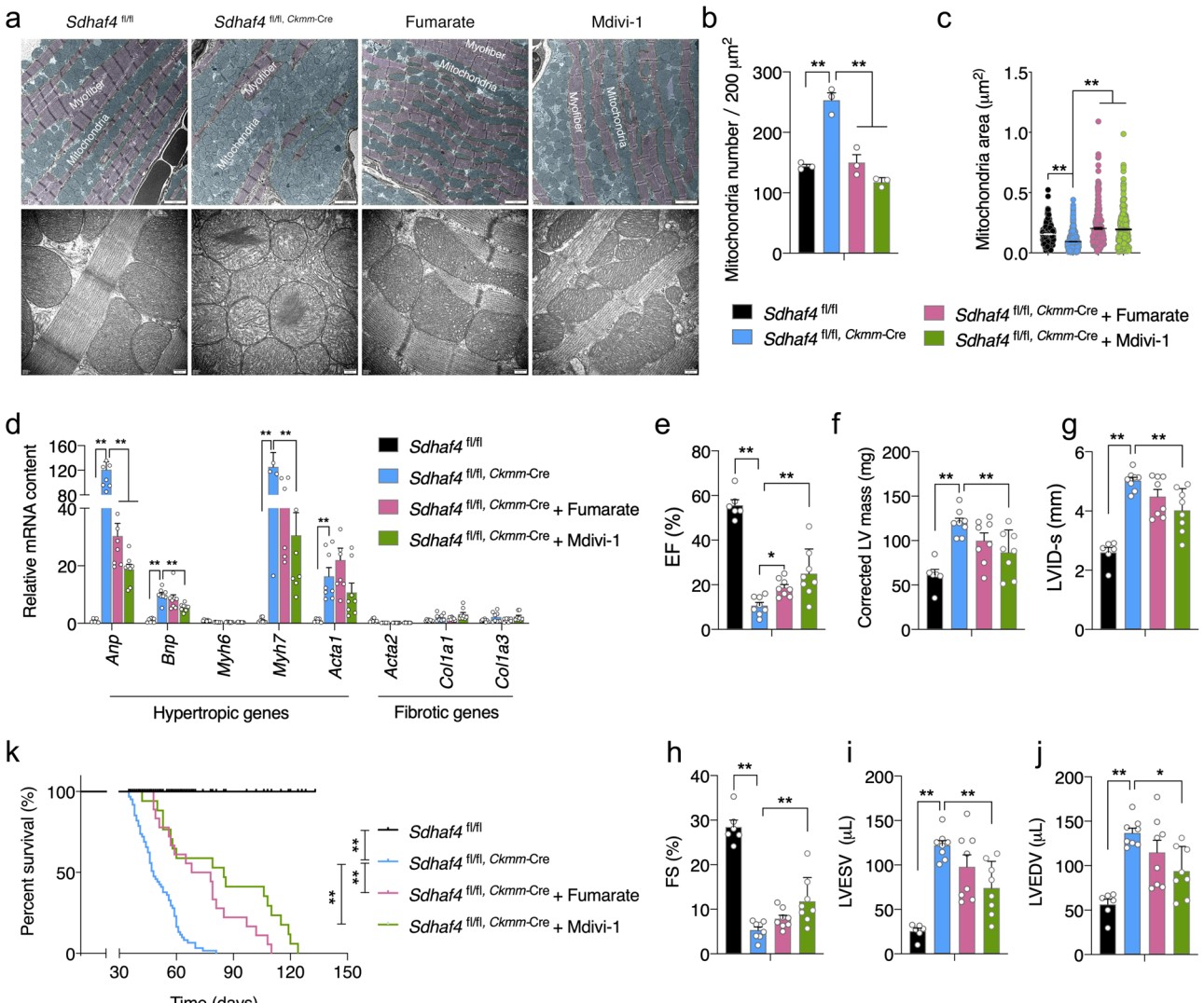

**Fig. 7 Inhibition of Drp1 or supplement with fumarate prolongs lifespan of mice with dilated cardiomyopathy. a–c** TEM analysis for left ventricles of hearts from WT, *Sdhaf4* mutant mice, mutant mice with fumarate or Mdivi-1 supplement for 4 weeks, representative images (**a**, scale bar, 1 μm and 200 nm respectively), mitochondrial number per section (**b**, $n = 3$, $P < 0.0001$), average mitochondrial area (**c**, $n = 293$ for WT, $n = 650$ for mutants, $n = 305$ for fumarate, $n = 230$ for Mdivi-1, $P < 0.0001$). **d** Relative expression of hypertrophic and fibrotic-associated genes in left ventricle of hearts of mice ($n = 8$, $P < 0.0001$). **e–j** Heart function measurements of WT, *Sdhaf4* mutants, *Sdhaf4* mutant mice with fumarate or Mdivi-1 supplement for 4 weeks: EF (**e**, $P < 0.0001$, $P = 0.012$), corrected LV mass (**f**, $P < 0.0001$), LVID-s (**g**, $P < 0.0001$), FS (**h**, $P < 0.0001$), LVESV (**i**, $P < 0.0001$), and LVEDV (**j**, $P < 0.0001$, $P = 0.011$), $n = 8$. **k** Postnatal survival curve for WT ($n = 25$ for both sex), *Sdhaf4* mutant mice ($n = 60$ for both sex), *Sdhaf4* mutant mice with daily fumarate supplement ($n = 18$ for both sex) and *Sdhaf4* mutant mice with daily Mdivi-1 supplement ($n = 18$ for both sex), $P < 0.0001$. Values are mean ± SEM, *$P < 0.05$, **$P < 0.01$. Statistical significance was determined by two-tailed Student's *t*-test (**b–j**). Log-rank Mantel-Cox testing was performed for survival analysis (**k**). Source data are provided in Source Data file. EF ejection fraction, LV left ventricular, LVID-s left ventricular internal dimension end-systolic, FS fraction shortening, LVESV left ventricular end-systolic volume, LVEDV left ventricular end- systolic volume.

the *Sdhaf4*<sup>*Ckmm*</sup> animals treated with fumarate, although the difference did not reach statistical significance (Fig. 7e–j). More importantly, lifetime treatment of weaned *Sdhaf4*<sup>*Ckmm*</sup> mice with fumarate or Mdivi-1 substantially increased their viability (Fig. 7k), indicating that the approaches targeting TCA metabolism or mitochondrial dynamics are effective methods to correct cardiac dysfunction.

**Direct fumarate supplementation improves cardiac function in MI mice.** Loss of fumarate, given that it is the direct product of SDH catalysis, may present as the key feature in the heart of *Sdhaf4*<sup>*Ckmm*</sup> mice. Notably, the downregulation of fumarate and an increase of succinate level was also reported in diseased hearts

including MI[12,31], indicating that fumarate-associated metabolic deregulation may be one of the key processes relevant to cardiac pathogenesis. To test the assumption, the MI mice were supplemented with fumarate before and after the surgery. The heart weight/bodyweight ratio was dramatically increased after MI and sufficiently decreased by fumarate supplement (Fig. 8a). Echocardiography was employed to assess cardiac function in control and MI mice with or without fumarate supplement (Supplementary Fig. 8b). The heart rates were comparable among the four groups during the measurement (Supplementary Fig. 8c). Intriguingly, the animals with fumarate supplement post MI surgery up to 7 days started to present significant improvement on cardiac function including corrected left ventricular (LV) mass (Fig. 8b), ejection fraction (EF%, Fig. 8c), FS% (Fig. 8d), left ventricular end-

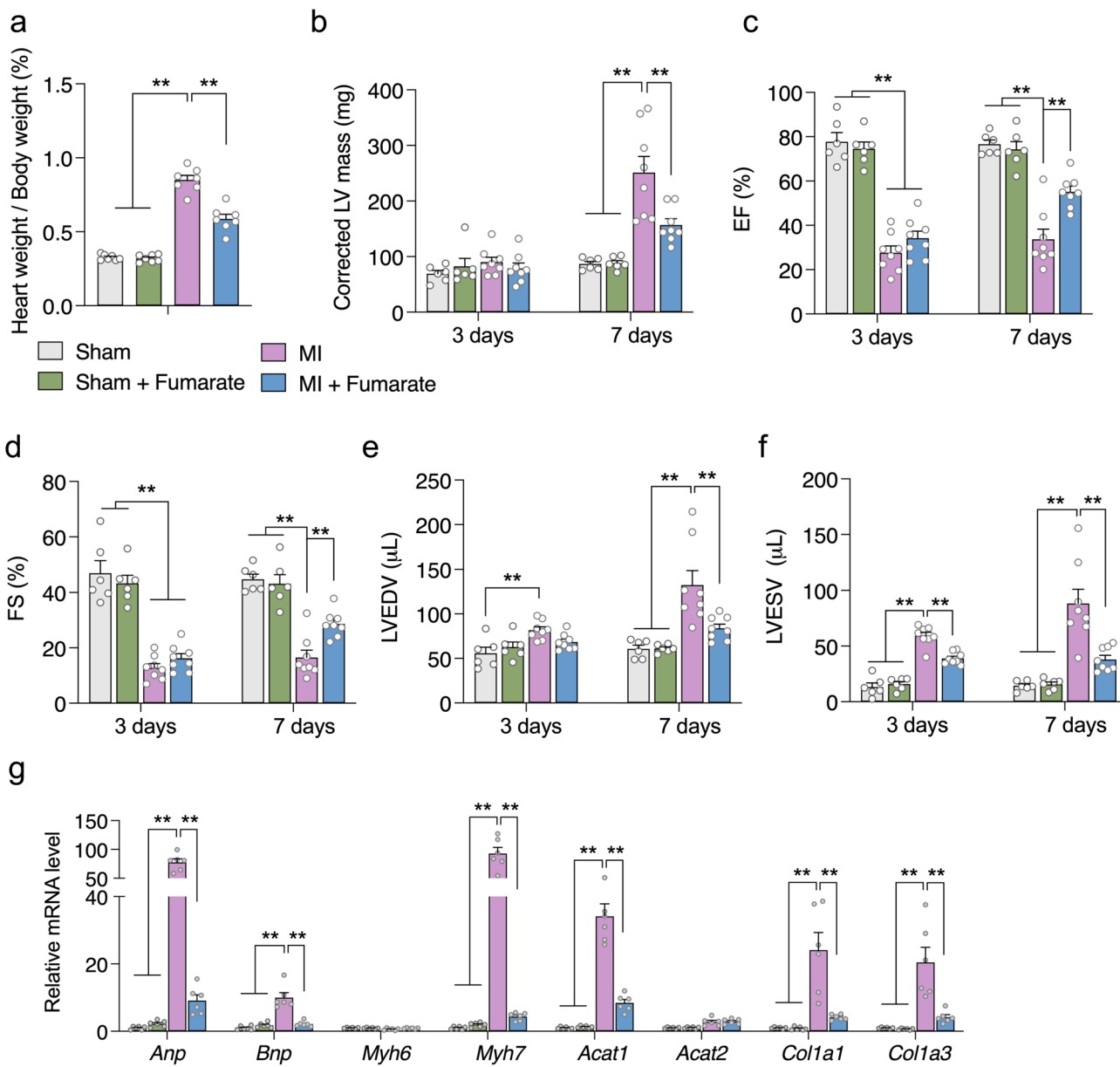

**Fig. 8 Fumarate supplement improves cardiac function in MI mice. a** Heart weight / bodyweight ratio of control and MI mice with or without fumarate supplement for 7 days, $n = 7$, $P < 0.0001$. **b–f** Heart function measurements of control and MI mice with or without fumarate supplement for 3 and 7 days ($n = 6$ for Sham and Sham + Fumarate groups, n = 8 for MI and MI + Fumarate groups): corrected LV mess (**b**, $P < 0.0001$), EF (**c**, $P < 0.0001$), FS (**d**, $P < 0.0001$), LVEDV (**e**, $P < 0.0001$), and LVESV (**f**, $P < 0.0001$). **g** Relative expressions of hypertrophic and fibrotic-associated genes in left ventricle of control and MI mice with or without fumarate supplement, $n = 6$, $P < 0.0001$. Values are mean ± SEM, ** $P < 0.01$. Statistical significance was determined by two-tailed Student's $t$-test (**a–g**). Source data are provided in Source Data file.

diastolic volume (LVEDV, Fig. 8e), and left ventricular end-systolic volume (LVESV, Fig. 8f) in MI mice. Further qPCR analysis of hypertrophic and fibrotic-associated genes in left ventricles consolidated the benefits of fumarate supplement in maintaining cardiac function after MI surgery (Fig. 8g). Collectively, the presented observations suggested that deficiency of SDHAF4-associated fumarate loss in MI condition plays vital role during the progression of cardiac dysfunction.

## Discussion
Various mitochondrial alterations have been implicated in heart diseases, yet due to the complexity and multiplicity of mitochondrial function, their pathological relevance and the

underlying mechanistic details remain elusive. Herein, downregulation of SDHAF4 represents one of the key cellular events during cardiac MI insult. Loss of *Sdhaf4* in cardiomyocytes led to a deregulated TCA cycle and abnormal mitochondrial dynamics and eventually resulted in the development of DCM and lethality in mice. Remarkably, supplementing both *Sdhaf4* mutants and MI animals with fumarate, an intermediate of the TCA cycle that was downregulated in the diseased hearts, substantially restored the cardiac function and improve viability of the mice. Deregulated TCA cycle and excessive mitochondrial division have been implicated in multiple human diseases, including cardiomyopathy. Herein, our results not only show that the decrease in fumarate levels and disordered mitochondrial dynamics are causes of cardiac defects but also suggest potential approaches to

target these events as promising therapeutic strategies for these diseases.

Mitochondrial respiratory complex II occupies a very unique place in mitochondrial function and metabolism. Although it has been implicated in various human diseases, the pathophysiological relevance of complex II deficiency in the development of cardiomyopathy remains elusive. We found that the abrogation of *Sdhaf4* in cardiac muscle resulted in a defect in complex II assembly and accelerated degradation of SDH subunits, which subsequently impaired mitochondrial function and led to DCM. A G555E mutation in the *SDHA* gene has been reported to cause DCM. Our study further highlights the key role of complex II in cardiac pathogenesis. More importantly, we predict that mutations in genes encoding complex II components, including SDHAF4, and/or dysregulation of their expression may be associated with cardiomyopathy in human patients.

Mitochondria-derived ROS has been suggested to play a central role in cardiac disease pathogenesis, and treatments targeting the mitochondrial ROS pool have been shown to be highly beneficial[5,32]. Mitochondrial ROS are mainly generated by complexes I and III, recently complex II has begun to be recognized as another source[33–35]. The loss of *Sdh8* (the orthologue of mammalian SDHAF4) in *Drosophila* was shown to cause excess ROS production[16]. Herein, we reported that SDHAF4 deficiency in cardiomyocytes impaired complex II activity and induced ROS overproduction and excess mitochondrial protein oxidation. Notably, these deregulations apparently occurred prior to the pathological alterations in the gross structure and morphology of the heart, thereby suggesting that complex II dysfunction may be an initial event in the pathological processes.

The formation of functional complex II requires stepwise interactions of assembly factors, including SDHAF1-4[36,37]. The current model suggests that SDHAF2 binds to SDHA, facilitating its flavination, which then binds to SDHAF4 for the dimerization of SDHA and SDHB. SDHAF1 and SDHAF3 assist in either the insertion or retention of [Fe-S] clusters within SDHB[14,36,37]. The subunits SDHC and SDHD are then incorporated into the holo-complex SDHA-SDHB through an as yet unclear process[38]. These factors act in the common pathway to regulate complex II assembly, yet the functional outcomes of deregulation on each individual subunit are not necessarily the same. These assembly factors may have diverse contributions to clinical pathology through distinct molecular networks, depending on the context, although their dysfunction results in consistent complex II deficiency. Currently, the clinical mutation of *SDHAF1* gene has been shown to be associated with infantile leukoencephalopathy[39–41]. *SDHAF2* mutations have been suggested to contribute to the development of paraganglioma[42,43]. A variant of *SDHAF3* appeared more prevalent in individuals with pheochromocytoma or paraganglioma[44]. However, the pathophysiological role of SDHAF4 protein in mammals currently remains elusive. Herein, we show that SDHAF4, compared to other complex II assembly factors, is more sensitive to cardiac insults. SDHAF4 deficiency in the heart led to severe cardiac abnormalities in mice. The pathogenesis involves several common molecular processes that also occur in disease models, including DCM, thereby suggesting that certain clinical DCM presentations may be associated with mutations or deregulation of *SDHAF4*.

As a direct substrate of SDH, succinate is converted to fumarate, which was present at decreased levels due to the excessive degradation of SDHA in *Sdhaf4* mutant mice. We assume that it is the initial metabolic incident in the development of DCM. In addition to the TCA cycle, fumarate is also involved in the arginine biosynthesis pathway. Fumarate and malate from the TCA cycle are utilized and undergo a series of catalytic reactions for the production of argininosuccinate, which is then converted

to arginine and fumarate by argininosuccinate lyase. This process links the mitochondrial TCA cycle to cytosolic amino acid metabolism[24,45,46]. We showed that decreased fumarate levels did not provoke compensatory activation of subsequent enzymes in the TCA cycle or related arginine biosynthesis pathways; instead, downregulation of these two processes, which may lead to severe metabolic inability, was observed. Amino acid metabolism might represent a compensatory pathway that fuels the TCA cycle through anaplerosis. In our metabolic profiling data, the abundances of phenylalanine and tyrosine, which are precursors for the TCA cycle anaplerotic entry point at the level of fumarate, were not significantly altered. Intriguingly, in our gene profiling data, two genes encoding key enzymes in the anaplerosis pathway, glutathione S-Transferase zeta 1(GSTZ1), which catalyzes the conversion of 4-maleylacetoacetate to 4-fumarylacetoacetate, and fumarylacetoacetate hydrolase (FAH), which is required for the hydrolysis of 4-fumarylacetoacetate to produce fumarate, were downregulated in the *Sdhaf4* mutant hearts (Supplementary Fig. 4a). Thus, anaplerosis of fumarate appeared to be attenuated in the mutant hearts and therefore was insufficient to compensate for the TCA cycle defects. Meanwhile, the decreased fumarate was also observed in diseased hearts including MI[12,31], and supplement of fumarate to both *Sdhaf4* mutants and MI mice could significantly improve their cardiac function indicating that fumarate-associated metabolic dysregulation may be one of the key processes relevant to cardiac pathogenesis.

In addition to metabolic disorders, we showed that Drp1-mediated excessive mitophagy is another important factor contributing to the development of DCM. Drp1 was discovered to be a vital regulator of mitochondrial fission and mitophagy[47,48]. The regulatory role of Drp1 in mitochondrial fission relies on its posttranslational modification status. Phosphorylation at Drp1$^{s616}$ is required for its pro-fission activity[49–52], whereas Drp1$^{s637}$ phosphorylation inhibits mitochondrial fission[53]. Consistent with these findings, we showed increased p-Drp1$^{s616}$ levels in the hearts of *Sdhaf4* mutant mice at the early postnatal stage along with lower p-Drp1$^{s637}$ levels. This result also supports the TEM observation of increased mitochondrial aggregates in the mutant cardiomyocytes. Moreover, we further showed that increased phosphorylation of Drp1$^{s616}$ was attributed to the activation of the ERK pathway, which apparently was induced by mitochondrial oxidative stress in the mutant hearts. We therefore propose that cardiac SDHAF4 deficiency induces progressive mitochondrial dysfunction manifested as metabolic dysfunction and excessive mitophagy, which eventually leads to the development of DCM.

In our study, targeting two major cellular pathways in *Sdhaf4* mutant mice by administering fumarate or Mdivi-1 improved mitochondrial quality, ameliorated cardiac dysfunction, and substantially restored the viability of the animals. However, the progression of DCM and subsequent heart failure were not completely reversed, thereby suggesting that the development of DCM results from the cumulative effects of complex cellular dysfunction involving additional pathological alterations. Gene expression is considered central to the progression of cardiac diseases. Our RNA-seq data showed that abrogation of *Sdhaf4* led to a drastic and broad deregulation of the transcriptome in the heart. As an assembly factor, SDHAF4 is very unlikely to modulate gene expression directly. However, loss of SDH activity in cardiac muscle led to succinate accumulation, which might subsequently inhibit alpha-ketoglutarate-dependent enzymes, including jumonji domain-containing histone demethylases (dioxygenase). Thus, *Sdhaf4* deficiency may trigger epigenetic alterations and consequently alter gene expression. We tested histone methylation in the mutant hearts. As shown in Supplementary Fig. 4, levels of methylated H3K4, H3K27, and H3K9

were indeed increased in the *Sdhaf4* mutant hearts. We assume that excess succinate may inhibit the activity of DNA demethylases and that DNA methylation is altered by the abrogation of *Sdhaf4*. Further investigations of these underlying processes will not only provide additional insights into the pathogenic mechanisms but also help to develop potential therapeutic interventions targeting these molecular events to treat heart diseases.

Taken together, the pathological involvement of SDHAF4 was studied in mice to determine its essential role in maintaining cardiac function. As a recently discovered complex II assembly factor, SDHAF4 expression was shown to be sensitive to stress. Deficiency of SDHAF4 disrupted complex II assembly and promoted the degradation of SDH units, thereby progressively leading to metabolic dysfunction and excess mitophagy, which eventually resulted in DCM and heart failure. Furthermore, we showed that interventions targeting mitochondrial metabolism and dynamics served as an effective strategy for improving cardiac function. The physiological evaluation of the SDHAF4 protein in mice provides more insights into complex II deficiency-associated DCM development and clinical practice.

## Methods

**Antibodies and reagents**. Antibodies against GAPDH (5174s), VDAC (4866), SDHA (11998s), TFAM (8076), Citrate Synthase (14309), ACO2 (6571), IDH2 (56439), OGDH (26865), Succinyl-CoA Synthetase (8071s), Fumarase (4567s), MDH2 (11908), Phospho-Drp1 (Ser616) (3455), Phospho-Drp1 (Ser637) (4867), Drp1 (8570), OPA1 (80471 s), LC3B (3868), AMPKα (5832), p-AMPKα (8208), SDHAF2 (45849), and Histone H3 (4499) were obtained from Cell Signaling Technology (Danvers, MA) and used at the dilution of 1:2000. Anti-mouse IgG (H + L), F(ab')2 Fragment (Alexa Fluor® 488 Conjugate) (4408) and Anti-rabbit IgG (H + L), F(ab')2 Fragment (Alexa Fluor® 555 Conjugate) (4413) were obtained from Cell Signaling Technology and used at dilution of 1:200. Antibodies against mitochondrial NDUFA9 (PA5-83598), UQCRC1 (459140), COX4 (MA5-15078), and ATP5A (459240) were purchased from Life Technologies (Waltham, MA) and used at the dilution of 1:2000. Antibodies against SDHB (178423), SDHAF4 (122196), SDHAF1 (185222), SDHC (155999), FIS1 (71498), and SDHD (189945) were purchased from Abcam (Cambridge, UK) and used at dilution of 1:2000. Antibody SDHAF3 (NBP2-14259), HIF1α (NB100-105) and PGC1α (NBP1-04676) were purchased from Novus Biologicals (Centennial, CO) and used at the dilution of 1:2000. Antibodies H3K4me1 (39635), H3K27me2 (39920), H3K9me2 (39041) and H3K9me3 (61013) were purchased from Active motif (Carlsbad, CA) and used at dilution of 1:1000. Antibodies ubiquitin (8017), MFN1 (166644), MFN2 (515647), and SQSTM1 (28359) were purchased from Santa Cruz Biotechnology (Santa Cruz, CA) and used at dilution of 1:1000. Protein oxidative modification detect kit (OxyBlot Protein Oxidation Detection Kit) (s7150) was purchased from Sigma-Aldrich (St. Louis, MO). Cell-permeable mitochondrial division Drp1 inhibitor Mdivi-1 (144589) was purchased from Abcam (Cambridge, UK). Cell culture medium was purchased from Life Technologies (Waltham, MA). Other reagents used in this study were purchased from Sigma (St. Louis, MO).

**Animals**. Cre/loxP system was employed to achieve tissue-specific transgenic mice. *Sdhaf4*-floxed mice (homozygous referred as *Sdhaf4*fl/fl, heterozygous referred as *Sdhaf4*fl/-) were generated by Beijing Biocytogen Co. Ltd. Briefly, two sgRNAs were designed to mediate the insertion of two loxP sites franking a ~3 kb chromosomal region (exon 1-2) at the *Sdhaf4* locus in the mouse genome. All the mice were in C57BL/6 J background and backcrossed at least seven generations. Ckmm-cre (No. 006475) mice and Mer-CreMer (Myh6-cre/Ers1*, No. 005657) were purchased from Jackson Laboratory. Ckmm-cre mice were crossed with *Sdhaf4*fl/fl mice to generate skeletal and cardiac muscle-specific knockout mice (*Sdhaf4*fl/fl, Ckmm-Cre, *Sdhaf4*fl/-, Ckmm-Cre referred as homozygous and heterozygous knockout, respectively). The littermates *Sdhaf4*fl/fl mice were used as control group. Similarly, cardiac myocytes-specific knockout mice were achieved through crossing *Sdhaf4*fl/fl mice with Mer-CreMer mice. Homologous recombination was achieved by intraperitoneally injecting of tamoxifen. The littermates *Sdhaf4*fl/fl, Mer-CreMer mice that were intraperitoneally injected with corn oil were used as control. For the intervention study, *Sdhaf4*fl/fl, Ckmm-Cre mice were daily injected with Mdivi-1 (Sigma-Aldrich, M0199) dissolved in DMSO at the dose of 50 mg/kg bodyweight, and *Sdhaf4*fl/fl, Ckmm-Cre mice injected with DMSO were used as control. The supplement of sodium fumarate dibasic (Sigma-Aldrich, F1506) to *Sdhaf4*fl/fl, Ckmm-Cre mice were through drinking water at the dose of 2 g/L. All the treatments were started at age of 4 weeks old, physiological and biochemical indicators were detected after 3 weeks of treatment. Survival rates were calculated through continuous supplement. Mice were fed in a temperature- and light/dark cycle-controlled animal room with free access to food and water. The protocol was approved by the Animal Care and Use Committee of the School of Life Science and Technology, Xi'an Jiaotong University (No.2017-0025). All investigations were carried out in terms of the United States Public Health Services Guide for the Care and Use of Laboratory Animals.

**Myocardial infarction**. Myocardial infarction (MI) of C57Bl/6 mice at age of 8 weeks was induced as previously described[17]. Briefly, general anesthesia was performed with 1-2% isoflurane and maintained on a ventilator. The heart was exposed to a left open thoracotomy, and the left anterior descending branch (LAD) was ligated with a 6-0 polyester thread 1 mm from the normal position of the left pinna tip. The surgical wound was sutured, and animals were allowed to recover. For the fumarate supplementation, the sodium fumarate dibasic (Sigma-Aldrich, F1506) was admitted to Sham and MI group at the dose of 2 g/L through drinking water 1 day prior to the surgery and lasted 7 days after the surgery.

**Echocardiography**. The mice were anaesthetized via the continuous inhalation of a mixture of isoflurane (1.5%) and oxygen. Cardiac ultrasound was performed using a Visual Sonics Vevo 770 Imaging System with a 30-MHz frequency transducer (VisualSonics Inc., Toronto, Ontario, Canada). M-mode echocardiographic images were recorded. LV mass, LV, EF, and FS were calculated according to the standard formulae.

**Histology**. Mouse hearts were collected, fixed with 4% paraformaldehyde, underwent tissue processing and embedded in paraffin. H/E, Masson and Sirius Red stainings of tissue sections (3–4 μm) were carried out using standard protocols.

**Electron microscopy**. Transmission electron microscopy (TEM) for morphological analysis was performed at the Instrument Analysis Center, Xi'an Jiaotong University, according to standard operating procedures. Briefly, fresh ventricular wall tissue was fixed in TEM fixing buffer (2.5% glutaraldehyde in phosphate buffer) overnight at room temperature. Ultrathin sections were mounted onto a copper grid and stained with uranyl acetate and lead citrate, imaged with a JEOL 1400 transmission electron microscope.

**Mitochondrial isolation and measurement of complex activities**. Mitochondria were extracted from the fresh left ventricular wall tissue following published protocol[54,55]. About 200 mg tissue was cut into small pieces with scissors and homogenized in ice-cold mitochondria extraction buffer (MEB) (225 mmol mannitol, 75 mmol sucrose, 1 mmol EGTA, 20 mmol HEPES-KOH and 0.5% free fatty acids-free BSA [pH 7.4]) using a polytron homogenizer (Kinematica, PT3100, Malters, Switzerland). The homogenate samples were centrifuged at $800 \times g$ for 10 min at 4 °C. The supernatant was collected to a new 1.5 ml conical tube and centrifuged at $12,000 \times g$ for 15 min at 4 °C. The supernatant was discarded, and the pellet was resuspended by MEB and centrifuged at $12,000 \times g$ for 15 min at 4 °C to wash the mitochondria twice, the last time washed by the MEB without BSA. Mitochondria were resuspended in MEB (without BSA). The concentration of the mitochondria was determined by the Compat-Able™ BCA Protein Assay Kit (Thermo Scientific™, No. 23229).

The activity of the complexes on ETC was performed according to methods previously described[56]. Briefly, complex I catalyzes the production of reducing CoQ, which can undergo quantitative non-enzyme-catalyzed chemical reaction with DCIP (2,6-Dichlorophenolindophenol) to make the color of reaction solution (50 mM Tris-HCl, pH8.1, 0.35% free fatty acids-free BSA, 1 μM antimycin A, 0.2 mM sodium azide, 50 μM CoQ, 50 μM DCIP, and 0.2 mM NADH to starting the reaction) containing DCIP gradually fade. The color fading rate of reaction solution was monitored to characterize the catalytic activity of complex I. The same principle was used to monitor the rate at which complex II catalyzes the production of reducing CoQ, and this rate characterized the catalytic activity of complex II with reaction solution (50 mM $K_3PO_4$, pH 7.8, 2 mM EDTA, 0.1% free fatty acids-free BSA, 3 μM rotenone, 1 μM antimycin A, 0.2 mM sodium azide, 200 μM ATP, 50 μM CoQ. 50 μM DCIP, and 10 mM sodium succinate to starting the reaction). The reaction rate of cytochrome *c* reduced by ubiquinol cytochrome *c* reductase was used to characterize the enzyme activity of complex III (reaction solution, 50 mM Tris-HCl, pH 7.8, 0.2 mM sodium azide, 0.05% tween-20, 0.01% free fatty acids-free BSA, 50 mM decylubiquinol, 50 μM cytochrome *c*). The rate at which reduced cytochrome *c* was oxidized by cytochrome *c* oxidase to characterize the enzyme activity of complex IV (reaction solution, 50 mM $K_3PO_4$, pH 7.0, 0.01% free fatty acids-free BSA, 0.2% tween-20, 50 μM reduced cytochrome *c* to starting the reaction). ATP synthase activity detection adopted the principle of enzyme cascade reaction that ATP synthase used the energy generated by succinic acid through successive enzymatic catalytic reaction, and then in the presence of ADP and inorganic phosphoric acid, the generated ATP was used by hexokinase to synthesize glucose-6-phosphate. The glucose-6-phosphate dehydrogenase subsequently converted NADP + to NADPH in the presence of glucose-6-phosphate, thus detecting the changes rate in the amount of NADPH to reflecting ATP synthase activity (reaction solution, 10 mM HEPES, pH8.0, 20 mM glucose, 3 mM $MgCl_2$, 0.75 mM $NADP^+$, 20 mM sodium succinate, 5 U hexokinase, 2.5 U glucose-6-phosphate dehydrogenase, 10 mM K2HPO4, and 1 mM ADP to starting the reaction).

**Oxygen consumption rate (OCR) assay**. Equal amount of isolated mitochon-drion were distributed in microplates (Seahorse Bioscience, Billerica, MA). After treating with mitochondrial inhibitors (1 μM antimycin A, 0.5 μM FCCP and 1 μM oligomycin), the OCR was detected using an extracellular flux analyzer (Seahorse Bioscience, Billerica, MA). Basal respiration represented the baseline of the oxygen consumption value before the injection of the mitochondrial inhibitors. Maximal respiration represented the maximum OCR value after injection of FCCP. Spare respiratory capacity was calculated by recording the OCR response to FCCP and dividing the value by the basal respiration. The actual value of OCR was adjusted according to the cellular protein concentration.

**ATP determination**. Relative ATP levels in cardiac tissues were analyzed following manufacture instruction of commercial ATP assay kit (Beyotime, S0026, China). Briefly, 500 μL lysis buffer from ATP detection kit was add to 50 mg fresh heart tissue and then ultrasonicated. The lysate was centrifuged at $12,000 \times g$ for 5 min at 4 °C. The supernatant was then transferred to a new sterile tube for ATP test.

**Cell culture**. Rat H9C2 cells (ATCC®CRL-1446™) and mouse C2C12 cells (ATCC® CRL-1772™) were cultured in growth medium consisting of DMEM supplemented with 10% fetal bovine serum (FBS and 1% Penicillin/Streptomycin). Cell cultures were maintained at 37 °C in a humidified atmosphere of 95% air and 5% $CO_2$. Cell medium was changed every 2 days.

**Plasmids**. Sdhaf4 shRNA were constructed targeting three different coding sequences. Sdhaf4 overexpression construct was generated by PCR amplification from cDNA derived from C2C12 cells. pLVX-Puro-mtKeima plasmid was kindly provided by Dr. Xin Pan (Institute of Basic Medical Sciences, National Center of Biomedical Analysis). Plasmid transfection was performed using Lipofectamine 3000 (Invitrogen, L3000150) according to the manufacturer's instructions.

**Metabolomics**. Ventricular wall tissue samples were delivered to Metabo-Profile R&D Laboratory (Shanghai, China) in dry ice for metabolomics analysis. Simply, the frozen samples were mixed with 25 mg of pre-chilled zirconium oxide beads and 10 μL of internal standard. Each aliquot of 50 μL of 50% pre-chilled methanol were added for automated homogenization (BB24, Next Advance, Inc., Averill Park, NY, USA). The supernatant was collected by high-speed centrifugation ($14,000 \times g$, 4 °C, 20 min). Each aliquot of 175 μL of pre-chilled methanol/chloro-form ($v/v = 3/1$) was added to the samples and fully mixed. High-speed cen-trifugation ($14,000 \times g$, 4 °C, 20 min) to collected supernatant, 200 μL to an autosampler vial and the residual supernatant was pooled to make quality control samples. The FreeZone freeze dryer (Labconco, Kansas City, MO) was employed to dry the samples. The dried sample was derivatized with 50 μL of methoxyamine (20 mg/mL in pyridine) at 30 °C for 2 h, followed by the addition of 50 μL of MSTFA (1% TMCS) containing FAMEs as retention indices at 37.5 °C for another 1 h using the sample preparation head. In parallel, the derivatized samples were injected with sample injection head after derivatization. Metabolites in the study samples were annotated with mammalian metabolite database JiaLib$^{TM}$ using a strict matching algorithm incorporated in XploreMET software (Version 2.0) that used both retention times and fragmentation patterns in the mass spectrum. Stu-dent t-test (T-test) was used to determine whether the two sets of data are sig-nificantly different or not. The calculated fold change of 1.5 or p-value of 0.05 was chosen for statistical significance, the raw data were provided in Supplementary data 2.

**Transcriptomics**. Fresh ventricular wall tissue samples were chopped in Trizol (Invitrogen, 15596018) and delivered to Novogene Co., Ltd., (Beijing, China) in dry ice for RNA-seq analysis. Briefly, mRNA was purified from total RNA using poly-T oligo-attached magnetic beads. Reference genome and gene model annotation files were downloaded from genome website directly. HTSeq v0.9.1 was used to count the reads numbers mapped to each gene. Genes with an adjusted P-value < 0.05 found by DESeq were assigned as differentially expressed. Corrected P-value of 0.005 and log2 (Fold change) of 1 were set as the threshold for significantly dif-ferential expression.

**Western blot**. The left ventricular wall tissue was homogenized in lysis buffer (20 mM Tris, pH 7.5, 150 mM NaCl, 1% Triton X-100 and 1 mM PMSF) sup-plemented with Protease Inhibitor Cocktail (Roche). The supernatant was collected by high-speed centrifugation ($12,000 \times g$) at 4 °C, using BCA method to measure concentration, and mixed with 4× SDS loading buffer (150 mM Tris-HCl, pH 6.8, 12% SDS, 30% glycerol, 0.02% bromophenol blue, 6% 2-mercaptoethanol). The samples were boiled for 10 min. Forty micrograms of protein from each sample was separated using 8%–15% gels by SDS-PAGE and transferred to a nitrocellulose membrane. After blocking with 5% nonfat milk in PBS (pH 7.4), the membranes were incubated with primary antibodies against at 4 °C overnight and visualized by chemiluminescent HRP substrate (Immun-Star HRP, BIO-RAD). Images were quantified using densitometric measurements by Clinx chemi analysis software (Clinx Science Instruments, Shanghai, China, Version 2.1.1.8).

**Immunoprecipitation**. Total proteins were collected from fresh tissues using western and immunoprecipitation lysis buffer (Beyotime, Jingsu, China). To avoid nonspecific binding, lysates were precleared with protein G Sepharose beads (GE Healthcare, NJ). Protein G Sepharose beads were subsequently removed via cen-trifugation for 10 min at 4 °C, and supernatants were incubated with appropriate antibodies for 1 h at 4 °C and then with fresh protein G Sepharose beads overnight at 4 °C with gentle rotation. Samples were finally washed three times with IP lysis buffer, resuspended in SDS sample buffer, boiled, and centrifuged for 5 min. Supernatants were further analyzed by western blot analysis.

**Protein carbonylation assay**. Fresh tissue homogenate was generated by fresh left ventricular wall in cell lysis buffer (20 mM Tris, pH 7.5, 150 mM NaCl, 1% Triton X-100). The homogenate samples were centrifuged at $800 \times g$ for 10 min at 4 °C. Collected the supernatant to a new 1.5 ml conical tube and centrifuged at $12,000 \times g$ for 15 min at 4 °C, collect the supernatant and pellet separately determined as cytoplasmic and mitochondrial proteins, using BCA method to unified concentration. Protein carbonyls in soluble proteins were assayed using an OxyBlot Protein Oxidation Detection Kit (Cell Biolabs, San Diego, CA). Protein carbonyls were labeled with 2,4-dinitrophenylhy-drazine and detected by western blot analysis. As a loading control, equal amounts of samples were subjected to 10% SDS-PAGE and stained with Coomassie brilliant blue.

**Blue native PAGE**. Blue native PAGE (BN-PAGE) testing was performed as previously described[57]. Briefly, mitochondria were extracted from the fresh left ventricular wall tissues and the concentration was unified by BCA method. Samples were mixed with native PAGE sample buffer to 1x final concentration and were loaded onto native PAGE gels alongside native marker standard and then the gels were run for 2 h to western blot analysis.

**mRNA extraction and qPCR**. Total RNA was isolated from cardiac tissues with TRIzol Reagent (Sigma-Aldrich, St. Louis, MO, Cat # T9424) and reverse-transcribed into cDNA with PrimeScript RT Master Mix (TaKaRa, DaLian, China, Cat # RR047A), followed by semi-quantitative real-time PCR with particular pri-mers. The $2^{-\Delta\Delta CT}$ method was used to analyze the data, and 18 S rRNA was adopted as a housekeeping gene to normalize the data. The results were presented as relative value to the control group. The primers adopted in real-time PCR were listed in Supplementary data 1.

**Statistical analysis**. Data were analyzed with GraphPad Prism software (V5 & V7) and presented as the mean ± standard error of the mean (SEM). Pairwise com-parisons were analyzed using two-tailed Student's t-test. Log-rank Mantel-Cox testing was performed for survival analysis. Other data were analyzed using one-way ANOVA with Tukey's multiple comparison test. In all cases, $P < 0.05$ was considered significant.

**Reporting summary**. Further information on research design is available in the Nature Research Reporting Summary linked to this article.

## Data availability

The data supporting the finding of this study are available within the manuscript and its supplementary information. The RNA-Seq data generated in this study have been deposited in NCBI Gene Expression Omnibus under accession code GSE163809. The metabolomics raw data generated in this study have been deposited in MetaboLights under accession code MTBLS4579. Transcriptomic data from human hearts generated by deep sequencing are available from NCBI Gene Expression Omnibus under accession code GSE135055. Source data are provided with this paper.

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

## Acknowledgements

The authors thank Dr. Xin Pan at Institute of Basic Medical Sciences, National Center of Biomedical Analysis for the kind gift of plasmid and Lei Chen, Shujun Han, Hua Li, Qiong Sun, Yan Wang, and Zhen Wang at the Center for Mitochondrial Biology and Medicine, Xi'an Jiaotong University for their technical assistance. The authors also thank the Instrument Analysis Center of Xi'an Jiaotong University for assistance with meta-bolomics and TEM analyses. This work was supported by the National Natural Science Foundation of China (32071154 to Z.F.; 31701025 to K.C.; 31771371 and 31970654 to J.D.; 81970227 to J.C., 31570777, and 32171102 to J. Liu), the Natural Science Foundation of Shaanxi (2018JZ3005 to J. Long), the Natural Science Basic Research Program of Shaanxi (2022JQ-767 to K.C.), Zhejiang Provincial Natural Science Foundation of China (LZ20H020001 to J.C.), the General Financial Grant from the China Postdoctoral Science Foundation (2021M692580 to K.C.).

## Author contributions

J. Liu and Z.F. conceived and supervised the study; Z.F., J.D., W.Z., F.G., J. Long, and J. Liu designed the experiments; X.W., X.Z., K.C., M.Z., and X.F. performed most experiments with the help of J.D., W.L., A.Z., H.L., X.Z., M.L., J.X., F.G., J.C.; and X.W., X.Z., K.C., and Z.F. analyzed the data; X.W., Z.F., and J. Liu wrote the manuscript.

## Competing interests

The authors declare no competing interests.
