## [Peer Review File · Nature Communications]

REVIEWER COMMENTS

Reviewer #1 (Remarks to the Author):

The MS by Wang et al on SDHAF4 is quite interesting and underscores the importance of mitochondrial function in determining metabolic output and cardiac contractility. The investigators noted the downregulation of the SDH assembly factor 4 in response to ischemic stress in the heart. Based on that observation they created a knockout mouse (restricted to cardiac and skeletal muscle) which exhibited progressive heart failure and early death. They demonstrated impaired assembly of SDHA and B and showed global mitochondria dysfunction, excessive mitochondrial fission, and metabolic derangements consistent with loss of a key enzyme in the TCA cycle. Systemic administration of fumarate or the fission inhibitor mdivi-1 partially attenuated cardiac dysfunction and extended survival. The study is novel and quite interesting. However, there are methodological problems and other concerns, as listed below:

Major Concerns:

- The study begins from the observation that SHDAF1, 3, and 4 are downregulated after MI in mice. They fail to indicate when after MI this happens, or how long it persists, or even whether protein levels are altered. I consider this rather thin justification for making a knockout mouse. It's not clear to me that the SDHAF4 KO sheds any light on post-MI heart failure or any other disease process other than the specific genetic defect. Metabolic changes after MI that would recapitulate what is seen in the KO would be supportive, as well as further demonstration that SDH fails to assemble. Additionally, it would be nice to close the loop by showing that fumarate supplementation can mitigate post-MI heart failure. Without relating this to a relevant human disease such as MI, this is merely a story of what happens when you knock out a mitochondrial assembly factor.
- Mitochondrial methods are rather problematic. Mitochondria were isolated in an EDTA-containing buffer, which will chelate Mg⁺⁺ needed for ATP complexation. They were then resuspended in PBS. High inorganic phosphate is not good for mitochondria (PMID: 15512800). Dounce homogenization also damages mitochondria, causing loss of cytochrome c, which would also affect respirometry (in both WT and KO), but differential effects can arise when mitochondria are fragile.
- As phosphate excess will result in elevated ROS production, the measurement of ROS damage (protein carbonylation) should be repeated using a more appropriate mitochondrial buffer.
- They performed respirometry but did not indicate what substrate was used. At 3 weeks of age, one might expect to see impaired respiration of a complex II substrate but normal respiration with complex I substrates.
- They show mito fragmentation—which can happen when a cell is energy-starved—but should measure AMPK phosphorylation, ATP/AMP, and other energetic parameters to provide appropriate information about the cell's energy balance.

- They mention mitophagy but only show a modest increase in LC3-II in KO mice or ARPE-19 cells. More specific mitophagy markers should be measured, as well as better measures of mitochondrial biogenesis which was mentioned but not really assessed. Given the profusion of mitochondria in Fig. 6a, it is also important to assess biogenesis.
- There is no good reason for using a retinal pigment epithelial line for these studies; a cardiac-related cell line or primary cardiomyocytes should be used.
- The metabolic imbalance driven by SDH deficiency may affect jumonji dioxygenases and broad gene expression. The paper would be more complete and less speculative (see line 200) if they measured histone methylation rather than simply inferring from pathway analysis that HIF-1a is involved and then attributing its activation to cardiac dysfunction.

Minor Concerns:

- Echo studies fail to report heart rate; low rates can compromise the reliability of the measurements, and anesthesia or hypothermia can affect HR. It is therefore an important parameter to provide in order to assess the validity of the measurements.
- There are numerous errors of spelling, punctuation, grammar, and syntax that will require extensive editing to make it readable and to convey meaning accurately. Some statements are subject to misinterpretation because of the confusing language.
- By and large, the number of replicates is appropriate and experimental data are of good quality.

Signed,

Roberta A. Gottlieb, MD

Reviewer #2 (Remarks to the Author):

Overall, this is a good study showing that knockout of *Sdhaf4* in the mouse induces hypertrophic cardiomyopathy associated with mitochondrial fission and respiratory dysfunction. Just a few comments:

1. Please comment on anaplerosis of fumarate from phenylalanine or tyrosine and why it is not sufficient in the KO mice
2. The hearts in figure 2 d (top panels) are upside down, please invert. Also the longitudinal sections do not reveal LV dilatation, just wall thickening, not consistent with the cross sections. Please explain in results section.
3. There are some grammatical/linguistic errors that need to be edited.

Reviewer #3 (Remarks to the Author):

Wang et al. are investigating the role of SDHAF4, an assembly factor of the succinate dehydrogenase (SDH) complex, in light of dilated cardiomyopathy (DCM). The association of succinate dehydrogenase deficiency and cardiac diseases is not new, however, the author show for the first time the specific pathogenesis in SDHAF4 deficient cardiac cells. For that purpose, the authors generated mouse models with cardiac loss of Sdhaf4, and characterised these phenotypically, on cellular, molecular and metabolic level.

My main criticism about the presented study is the lack of clinical data on human tissues. Is there publicly available expression data for tissues from DCM patients versus healthy hearts? qPCR, immunohistochemistry (or western blot) should be employed to actually show that SDHAF4 deficiency is observed clinically. In relation to this, I could not follow the narrative of the introduction, why the authors decided to generate models with Sdhaf4 knockout and not with any other assembly factor or SDH complex. Mutations in SDHA have actually been shown to clinically cause DCM in humans, but for SDHAF4 nothing is known.

In addition, I advise to incorporate the following considerations for improving the current manuscript:

1. I strongly advise for a native English speaker with science background to edit the manuscript. Many parts of the text are reasonable and understandable, however, there are sentences that are weird to read and don't always make sense. Here a few examples (the list is not complete):

Line 60-63: Why does being a "cellular powerhouse" precludes the emergence of being an "integral player" in a number of processes?

Line 68/69: sentence structure after "however" wrong

Line 270/71: I do not understand what the conclusion is stated here

Line 93/94; Line 235/36 , Line 279/80, etc.

2. Several times in the manuscript, SDHAF4 is referred to as “newly identified”. As referenced by the authors, the paper describing its function was published in 2014. I would rephrase to “most recently identified” where appropriate.

3. Line 146/47: Please rephrase the conclusion for this paragraph. What “complicated multiple biological defects”? This sentence does not say much to me. This is the only point in the manuscript that the hypoxia-inducible factor (HIF) pathway is mentioned, but what significance it has is not explained.

4. Following up on point 3, it is well known that succinate is a so called oncometabolite that can inhibit alpha-ketoglutarate dependent enzymes. The authors show in Fig. 4c significant increases of succinate, but are these sufficient to inhibit alpha-ketoglutarate dependent enzymes, such as prolyl hydroxylases or DNA demethylase? In my opinion this is a missing piece in the presented characterisation of loss of function effects of SDHAF4. The expression analysis points to an upregulation of HIF signalling, but can HIF1/2alpha stabilisation be shown on protein level? Do the SDHAF4 deficient hearts possess a hypermethylated DNA profile at 3 weeks or 8 weeks? In light of the incomplete reversal of pathological effects by fumarate and mdivi, changes in global methylation may be responsible, since these may be only partially or only slowly reversible. Please also add to the discussion.

5. Line 240-243: Please rephrase the description of the results to better reflect Fig. 6. Treatments only partially reversed the different parameters and did not reach wild type levels. In many cases only mdivi treatment reaches statistical significance.

6. Fig 6k: Please add the statistics to the survival graph. Is the effect significant? Connected to this result. line 330 in the discussion should be edited, either say “significantly” if true or I just remove “efficiently”.

7. Methods: Details about used antibodies are missing. For statistical analysis of more than two groups it states “one way”? Do you mean one-way ANOVA? What posthoc tests did you use? Please also add the statistics concerning the survival analysis.

8. Fig. S1a: Please add the expression levels for heterozygous animals fl/- Ckmm-Cre

9. Fig. S3: Legends and axes titles are too small, higher quality image required

10. Line 281/82: SDHAF2 is a well-established susceptibility gene for paragangliomas, SDHAF3 not due to only one publication on SDHAF3 in pheochromocytoma/ paraganglioma.

11. Line 337: abbreviation MI stress?

12. Discussion: Related to my main critical point, there are to statements in the discussion that should be reconsidered or expanded.

Line 290/91: I agree that there might be the possibility that mutations in SDHAF4 play a role in DCM in humans; however, the authors did not present data showing this. They showed that loss of SDHAF4 function in mice leads to DCM, whether mutations in the actual gene or other mechanisms suppressing gene transcription or translation are responsible in the human is open.

Line 339/40: I don't agree that the presented study actually infers a clinical need for evaluation of SDHAF4 mutations in patients with DCM. The presented results surely warrant further investigation into the topic, where all SDH related genes should be investigated, not only SDHAF4.

Susan Richter

Response to Reviewer Comments

Reviewer #1 (Remarks to the Author):

The MS by Wang et al on SDHAF4 is quite interesting and underscores the importance of mitochondrial function in determining metabolic output and cardiac contractility. The investigators noted the downregulation of the SDH assembly factor 4 in response to ischemic stress in the heart. Based on that observation they created a knockout mouse (restricted to cardiac and skeletal muscle) which exhibited progressive heart failure and early death. They demonstrated impaired assembly of SDHA and B and showed global mitochondria dysfunction, excessive mitochondrial fission, and metabolic derangements consistent with loss of a key enzyme in the TCA cycle. Systemic administration of fumarate or the fission inhibitor mdivi-1 partially attenuated cardiac dysfunction and extended survival. The study is novel and quite interesting. However, there are methodological problems and other concerns, as listed below:

Major Concerns:

- The study begins from the observation that SHDAF1, 3, and 4 are downregulated after MI in mice. They fail to indicate when after MI this happens, or how long it persists, or even whether protein levels are altered. I consider this rather thin justification for making a knockout mouse. It's not clear to me that the SDHAF4 KO sheds any light on post-MI heart failure or any other disease process other than the specific genetic defect. Metabolic changes after MI that would recapitulate what is seen in the KO would be supportive, as well as further demonstration that SDH fails to assemble. Additionally, it would be nice to close the loop by showing that fumarate supplementation can mitigate post-MI heart failure. Without relating this to a relevant human disease such as MI, this is merely a story of what happens when you knock out a mitochondrial assembly factor.

Response: Thanks for the suggestions. These are really good points.

We re-performed MI surgery in mice and harvested the cardiac muscle samples at the different time points, 3, 14 and 28 days (MI_3d, MI_14d and MI_28d) after the surgery. We assessed expression of SDHAF 1, 2, 3 and 4 in these samples (border region of MI). The mRNA levels of SDHAF1, 3 and 4 were decreased in MI samples, while only decrease of SDHAF4 was statistically significant, and at MI_3d, the mRNA level of SDHAF4 was reduced by over threefold, and was about 50% downregulated at MI_14d and MI_28d (Fig. 1a). We further examined the protein abundance of these SDH assembly factors by western blot, IHC and IF analysis, and found that SDHAF4 was the one consistently showing the pattern of downregulation (Fig. 1b-d, Fig. S1b-e). We also surveyed the public available datasets of clinical samples (GSE135055)¹. Remarkably, decreased expression of SDHAF4 in the hearts appears to be closely associated with human cardiac disorders, including DCM (dilated cardiomyopathy) and MCD (microvascular coronary disease) (Fig. 1h). We have included these results and the interpretation in the revised manuscript.

We analyzed the published metabolic profiling datasets of different MI models². However, we did not find the significant overlap with our metabolomics data in *Sdhaf4*-KO hearts. Cellular metabolism is very sensitive to various stimuli and signal cues. The post-MI remodeling

involves multiple complex molecular, cellular and interstitial changes, resulting in general perturbations of homeostasis, with many metabolic pathways affected. We therefore do not take the position to assume that knockout of *Sdhaf4* will globally recapitulate the metabolic profiles in MI models.

In the revised manuscript, we compared the differentially expressed gene (DEG) list in *Sdhaf4*-KO hearts with previously reported gene profiling data of MI (MI-3d) model³. Intriguingly, expression of 31% (749/2420) of the upregulated genes in MI was increased in *Sdhaf4*-KO hearts. For the down-regulated genes in MI model, 39.53% (1065/2694) of them also displayed reduced expression in *Sdhaf4*-KO hearts (Fig. S4b). These results suggest that there are similar or common pathologic processes occurring in the two models, and the *Sdhaf4*-KO is relevant to the diseases. We have included these results and the interpretation in the revised manuscript.

It is a good point to test if fumarate supplementation can mitigate post-MI heart failure. Fumarate has been proposed to be cardiac protective via activation of the Nrf2 antioxidant pathway. The observations may call for follow up studies to address them mechanistically. It will be interesting to comprehensively characterize the effects of fumarate in various cardiac disease models, including MI, in the future. This will offer important insights into the therapeutic interventions for heart diseases.

- Mitochondrial methods are rather problematic. Mitochondria were isolated in an EDTA-containing buffer, which will chelate Mg⁺⁺ needed for ATP complexation. They were then resuspended in PBS. High inorganic phosphate is not good for mitochondria (PMID: 15512800). Dounce homogenization also damages mitochondria, causing loss of cytochrome c, which would also affect respirometry (in both WT and KO), but differential effects can arise when mitochondria are fragile.

Response: Thanks for the suggestions. We have followed the suggestions and repeated the experiments with modified buffer for mitochondria isolation and analysis following previous published studies^{4,5}. The new analysis presents consistent observation with previous one. We have updated the data and method in the revised manuscript (Fig. 4i, Fig. S5e).

- As phosphate excess will result in elevated ROS production, the measurement of ROS damage (protein carbonylation) should be repeated using a more appropriate mitochondrial buffer.

Response: We have followed the suggestion and repeated the experiments with standard mitochondrial buffer. As shown in Fig S5, protein carbonylation signal was consistently increased in KO hearts. We have updated the new data in the revised manuscript (Fig. S5f, g).

- They performed respirometry but did not indicate what substrate was used. At 3 weeks of age, one might expect to see impaired respiration of a complex II substrate but normal respiration with complex I substrates.

Response: Complex I catalyzes the production of reducing CoQ which can undergo quantitative non-enzyme-catalyzed chemical reaction with DCIP (2,6-Dichlorophenolindophenol) to make the color of reaction solution (50 mM Tris-HCl (pH8.1), 0.35% free fatty acids-free BSA, 1 μM antimycin A, 0.2 mM sodium azide, 50 μM CoQ, 50 μM DCIP, and 0.2 mM NADH to starting the reaction) containing DCIP gradually fade. The color

fading rate of reaction solution is monitored to characterize the catalytic activity of complex I. The same principle is used to monitor the rate at which complex II catalyzes the production of reducing CoQ, and this rate characterized the catalytic activity of complex II. FMN and FAD do not directly participate in DCIP reaction, but CoQ acts as intermediate electron carrier between them (reaction solution, 50 mM K₃PO₄ (pH7.8), 2 mM EDTA, 0.1% free fatty acids-free BSA, 3 μM rotenone, 1 μM antimycin A, 0.2 mM sodium azide, 200 μM ATP, 50 μM CoQ, 50 μM DCIP, and 10 mM sodium succinate to starting the reaction). We have updated the detailed method in revised manuscript.

- They show mito fragmentation—which can happen when a cell is energy-starved—but should measure AMPK phosphorylation, ATP/AMP, and other energetic parameters to provide appropriate information about the cell's energy balance.

Response: Thanks for the suggestions. We have followed the suggestions and measured the relative ATP level and AMPK phosphorylation. As shown in Fig. 4k, the relative ATP level remained unchanged 3 weeks after birth, although was slightly decreased in the hearts of 8-week-old *Sdhaf4*-KO mice. Neither total AMPK abundance nor phosphorylated AMPK level was altered in the cardiac muscle samples harvested from 8-week old *Sdhaf4*-KO mice (Fig. 6l). We have included these results in the revised manuscript.

- They mention mitophagy but only show a modest increase in LC3-II in KO mice or ARPE-19 cells. More specific mitophagy markers should be measured, as well as better measures of mitochondrial biogenesis which was mentioned but not really assessed. Given the profusion of mitochondria in Fig. 6a, it is also important to assess biogenesis.

Response: Thanks for the suggestions. In addition to the western blot experiment for LC3-II, we also assessed mitophagy in H9C2 rat cardiomyocyte cell line by using a fluorescent reporter (mt-Keima)-based approach. Keima is a lysosomal proteases-resistant, dual-excitation ratiometric fluorescent protein, which is sensitive to changes in pH. Fusion with mitochondrial signal peptide enables the protein (mt-Keima) to be localized into mitochondria, where normally the pH value is 8.0, and the short-wavelength excitation of Keima predominates. Once the mitochondria are uptaken by the acidic lysosome (pH4.5), mt-Keima will undergo a gradual shift to the longer-wavelength excitation. The mt-Keima reporter therefore allows us to more readily monitor mitophagy in the cells and has been widely used as a readout of mitophagy⁶. As shown in Fig. 6q, knockdown of SDHAF4 increased mitophagosome formation. We have included the description of mt-Keima in the revised manuscript.

We also tested if the increased numbers of mitochondria in the KO cardiomyocytes is potentially due to alteration of mitochondrial biogenesis. However, qPCR analysis demonstrated that relative abundance of mtDNA was decreased (Fig. 6k), instead of being up-regulated in mutant cardiac muscle. We further measured the levels of regulators of mitochondrial biogenesis, and found their expression was reduced in the KO mice (Fig. 6l, m). It is therefore unlikely that the increased quantity of the organelle is resulted from the change in mitochondrial biogenesis. We have included the updated results as well as the interpretation in the revised manuscript.

- There is no good reason for using a retinal pigment epithelial line for these studies; a cardiac-

related cell line or primary cardiomyocytes should be used.

Response: Thanks for the suggestions. We have followed the suggestions and repeated the experiments with H9C2 rat cardiomyocyte cell line. The similar results were obtained in H9C2 cells, and were included in the revised manuscript (Fig. 6q, Fig. S7a, b).

- The metabolic imbalance driven by SDH deficiency may affect jumonji dioxygenases and broad gene expression. The paper would be more complete and less speculative (see line 200) if they measured histone methylation rather than simply inferring from pathway analysis that HIF-1a is involved and then attributing its activation to cardiac dysfunction.

Response: Thanks for the suggestions. Loss of SDH activity can lead to succinate accumulation, which in turn can inhibit alpha-ketoglutarate-dependent jumonji domain-containing histone demethylases (dioxygenase). We have followed the suggestions and tested histone methylation in the KO hearts. As shown in Fig. S4c-d, methylation of H3K4, H3K27 and H3K9 were upregulated in the KO hearts. This is consistent with our metabolic profiling data showing succinate accumulation in mutants. SDHAF4 is an assembly factor, which is very unlikely to directly regulate gene expression. We speculate that the histone modification may participate in mediating the gene regulation in the KO hearts. We have included the interpretation in the revised manuscript.

Minor Concerns:

- Echo studies fail to report heart rate; low rates can compromise the reliability of the measurements, and anesthesia or hypothermia can affect HR. It is therefore an important parameter to provide in order to assess the validity of the measurements.

Response: Thanks for the suggestion. We have included the ECGs and heart rate data in sup Fig. S2f-g. As shown in the results, knockout of *Sdhaf4* did not significantly alter the ECG pattern or the heart rates.

- There are numerous errors of spelling, punctuation, grammar, and syntax that will require extensive editing to make it readable and to convey meaning accurately. Some statements are subject to misinterpretation because of the confusing language.

Response: Thanks for the suggestion. We have carefully gone through and re-edited the manuscript with the help of professional editors at American Journal Experts.

- By and large, the number of replicates is appropriate and experimental data are of good quality.

Response: We thank the reviewer for the favorable comments.

Signed,

Roberta A. Gottlieb, MD

Reviewer #2 (Remarks to the Author):

Overall, this is a good study showing that knockout of *Sdhaf4* in the mouse induces

hypertrophic cardiomyopathy associated with mitochondrial fission and respiratory dysfunction. Just a few comments:

1. Please comment on anaplerosis of fumarate from phenylalanine or tyrosine and why it is not sufficient in the KO mice

Response: This is a very good point. Amino acid metabolism could represent a compensatory pathway that could fuel the TCA cycle by anaplerosis. We found in our metabolic profiling data that the abundances of phenylalanine and tyrosine, which are precursors for the TCA cycle anaplerotic entry point at the level of fumarate, were not significantly altered. Intriguingly, in our gene profiling data, two genes encoding the key enzymes in the anaplerosis pathway, GSTZ1, which catalyzes the conversion of 4-maleylacetoacetate to 4-fumarylacetoacetate, and FAH, required for the hydrolysis of 4-fumarylacetoacetate to produce fumarate, were down-regulated in the KO hearts (Fig. S4a). These results indicate that anaplerosis of fumarate appeared to be attenuated in the mutant hearts, and therefore was insufficient to compensate the TCA defects. We have included the interpretation in the discussion of the revised manuscript.

2. The hearts in figure 2 d (top panels) are upside down, please invert. Also the longitudinal sections do not reveal LV dilatation, just wall thickening, not consistent with the cross sections. Please explain in results section.

Response: Following the suggestions, we have inverted the pictures in the revised manuscript. The longitudinal sections (top panels) were from the hearts of mice at the age of 4 weeks after birth. At this age, only cardiac hypertrophy was seen. The mutant hearts underwent cardiac remodeling and progressive dilation. At the latter stage (~6 weeks after birth), severe dilations of ventricular chambers were observed. We have included the description in the results section of the revised manuscript.

3. There are some grammatical/linguistic errors that need to be edited.

Response: Thanks for the suggestion. We have carefully gone through and re-edited the manuscript with the help of professional editors at American Journal Experts.

Reviewer #3 (Remarks to the Author):

Wang et al. are investigating the role of SDHAF4, an assembly factor of the succinate dehydrogenase (SDH) complex, in light of dilated cardiomyopathy (DCM). The association of succinate dehydrogenase deficiency and cardiac diseases is not new, however, the author show for the first time the specific pathogenesis in SDHAF4 deficient cardiac cells. For that purpose, the authors generated mouse models with cardiac loss of Sdhaf4, and characterised these phenotypically, on cellular, molecular and metabolic level.

My main criticism about the presented study is the lack of clinical data on human tissues. Is there publicly available expression data for tissues from DCM patients versus healthy hearts? qPCR, immunohistochemistry (or western blot) should be employed to actually show that SDHAF4 deficiency is observed clinically. In relation to this, I could not follow the narrative of the introduction, why the authors decided to generate models with Sdhaf4 knockout and not

with any other assembly factor or SDH complex. Mutations in SDHA have actually been shown to clinically cause DCM in humans, but for SDHAF4 nothing is known.

Response: Thanks for the suggestions.

We could not have clinical results at this point due to the limited availability of clinical samples, but such clinical studies are warranted in the future. In the current study we have included both MI mouse model and public datasets of clinical samples (GSE135055), and found significant association of SDHAF4 expression with cardiac pathology.

We re-performed MI surgery in mice and harvested the cardiac muscle samples at the different time points, 3, 14 and 28 days (MI_3d, MI_14d and MI_28d) after the surgery. We assessed expression of SDHAF 1, 2, 3 and 4 in these samples (border region of MI). The mRNA levels of SDHAF1, 3 and 4 were decreased in MI samples, while only decrease of SDHAF4 was statistically significant, and at MI_3d, the mRNA level of SDHAF4 was reduced by over threefold, and was about 50% downregulated at MI_14d and MI_28d (Fig. 1a). We further examined the protein abundance of these SDH assembly factors by western blot, IHC and IF analysis, and found that SDHAF4 was the one consistently showing the pattern of down-regulation (Fig. 1b-d, Fig. S1b-e). We also surveyed the public available datasets of clinical samples (GSE135055)¹. Remarkably, decreased expression of SDHAF4 in the hearts appears to be closely associated with human cardiac disorders, including DCM (dilated cardiomyopathy) and MCD (microvascular coronary disease) (Fig. 1h). We have included these results and the interpretation in the revised manuscript.

We also described the rationale of generating models with *Sdhaf4* knockout in the revised manuscript. Our results indicate that the expression of SDHAF4 is more sensitive to the cardiac insults, than that of other SDH assembly factors. We speculate that SDHAF4 may be a unique factor of complex II and play important roles in regulating cardiac homeostasis. Study of SDHAF4 may offer one entry point for us to further assess the involvement of complex II in cardiac remodeling and pathogenesis.

In addition, I advise to incorporate the following considerations for improving the current manuscript:

1. I strongly advise for a native English speaker with science background to edit the manuscript. Many parts of the text are reasonable and understandable, however, there are sentences that are weird to read and don't always make sense. Here a few examples (the list is not complete):

Line 60-63: Why does being a “cellular powerhouse” precludes the emergence of being an “integral player” in a number of processes?

Line 68/69: sentence structure after “however” wrong

Line 270/71: I do not understand what the conclusion is stated here

Line 93/94; Line 235/36 , Line 279/80, etc.

Response: We appreciate reviewer's suggestions and careful check. We have carefully gone through and re-edited the manuscript with the help of professional editors at American Journal Experts.

2. Several times in the manuscript, SDHAF4 is referred to as “newly identified”. As referenced by the authors, the paper describing its function was published in 2014. I would rephrase to “most recently identified” where appropriate.

Response: Thanks for the suggestions. We have rephrased the sentence in the revised manuscript.

3. Line 146/47: Please rephrase the conclusion for this paragraph. What “complicated multiple biological defects”? This sentence does not say much to me. This is the only point in the manuscript that the hypoxia-inducible factor (HIF) pathway is mentioned, but what significance it has is not explained.

Response: Thanks for the suggestions. We have rephrased the conclusions of the RNA-seq results. HIF signaling can be induced by cardiac dysfunction, and its activation has been observed in various cardiomyopathies, including ischemic heart disease, dilated cardiomyopathy and heart failure⁷. Previous studies have also demonstrated that HIF pathway was cardio-protective in various cardiac disorders⁸. Induction of HIF signaling therefore may act as one of the adaptive responses in the hearts of the mutant animals. We have included more detailed interpretation about HIF signaling in the revised manuscript.

4. Following up on point 3, it is well known that succinate is a so called oncometabolite that can inhibit alpha-ketoglutarate dependent enzymes. The authors show in Fig. 4c significant increases of succinate, but are these sufficient to inhibit alpha-ketoglutarate dependent enzymes, such as prolyl hydroxylases or DNA demethylase? In my opinion this is a missing piece in the presented characterisation of loss of function effects of SDHAF4. The expression analysis points to an upregulation of HIF signalling, but can HIF1/2alpha stabilisation be shown on protein level? Do the SDHAF4 deficient hearts possess a hypermethylated DNA profile at 3 weeks or 8 weeks? In light of the incomplete reversal of pathological effects by fumarate and mdivi, changes in global methylation may be responsible, since these may be only partially or only slowly reversible. Please also add to the discussion.

Response: Thanks for the questions. These are very good points. Loss of SDH activity can lead to succinate accumulation, which in turn can inhibit alpha-ketoglutarate-dependent enzymes, including jumonji domain-containing histone demethylases (dioxygenase). We tested histone methylation in the KO hearts. As shown in Fig. S4c-d, methylation of H3K4, H3K27 and H3K9 were upregulated in the KO hearts. This is consistent with our metabolic profiling data showing succinate accumulation in mutants. We agree with the reviewer that the excessive succinate can also inhibit the activity of DNA demethylases, and DNA methylation can be altered by abrogation of *Sdhaf4*, as well.

We also assessed the protein level of HIF and found they were up-regulated in the KO hearts (Fig. S4c-d). It may be resulted from insufficient oxygen supply caused by the defects in cardiac contraction and the abnormalities in myocardial blood flow. It also can be attributed to the excessive oxygen demand due to the increased ventricle cavity size. Previous studies have demonstrated that HIF pathway was cardio-protective in various cardiac disorders. Activation of HIF signaling therefore may partially act as one of the adaptive responses in the hearts of the mutant animals.

As an assembly factor, SDHAF4 is very unlikely to directly modulate gene expression, whereas the epigenetic factors, and also the HIF-1 signaling, may play key roles in mediating the regulatory effects. Given that fumarate and mdivi only partially repaired the defects in the mutants, it is possible that changes in histone and/or DNA methylation may be involved in and

be responsible for the pathogenesis, and the incomplete rescue may be ascribed to the altered epigenetic modifications. If so, in the future it will be interesting to systemically characterize DNA methylation (or hydroxymethylation) and histone modifications in the heart, and such in-depth analysis may offer more insights into the mechanisms underlying the pathogenesis. We have included the points and updated data in the revised manuscript.

5. Line 240-243: Please rephrase the description of the results to better reflect Fig. 6. Treatments only partially reversed the different parameters and did not reach wild type levels. In many cases only mdivi treatment reaches statistical significance.

Response: We agree with the reviewer that the treatments only partially rescued the defects. In the revised manuscript, we have rephrased the sentences: "...Echocardiography measurements showed that supplementation of Mdivi-1 significantly improved the heart function of the KO mice. A slight increase in fractional shortening (FS%) value was observed in the mutant animals treated with fumarate, although it did not reach statistical significance. More importantly, a lifetime intervention with fumarate or Mdivi-1 to weaned *Sdhaf4*^{Ckmm} mice increased their viability..."

6. Fig 6k: Please add the statistics to the survival graph. Is the effect significant? Connected to this result. line 330 in the discussion should be edited, either say "significantly" if true or I just remove "efficiently".

Response: Thanks for the suggestion. We have included *P*-value to the survival graph and rephrased the sentences in the revised manuscript.

7. Methods: Details about used antibodies are missing. For statistical analysis of more than two groups it states "one way"? Do you mean one-way ANOVA? What posthoc tests did you use? Please also add the statistics concerning the survival analysis.

Response: We have included information as suggested in the revised manuscript.

8. Fig. S1a: Please add the expression levels for heterozygous animals fl/- Ckmm-Cre

Response: The expression level for heterozygous animals have been updated in the revised figures (Fig. S2a).

9. Fig. S3: Legends and axes titles are too small, higher quality image required

Response: We have supplied higher quality image for the figure, they may, however, be converted to lower resolution PDF file during the submission.

10. Line 281/82: SDHAF2 is a well-established susceptibility gene for paragangliomas, SDHAF3 not due to only one publication on SDHAF3 in pheochromocytoma/ paraganglioma.

Response: Thanks for the suggestions. In the revised manuscript, we have rephrased the sentence as below: "A variant of SDHAF3 appeared more prevalent in individuals with pheochromocytoma or paraganglioma."

11. Line 337: abbreviation MI stress?

Response: We have included the full phrase (myocardial infarction (MI)) of the abbreviation.

12. Discussion: Related to my main critical point, there are to statements in the discussion that should be reconsidered or expanded.

Line 290/91: I agree that there might be the possibility that mutations in SDHAF4 play a role in DCM in humans; however, the authors did not present data showing this. They showed that loss of SDHAF4 function in mice leads to DCM, whether mutations in the actual gene or other mechanisms suppressing gene transcription or translation are responsible in the human is open.

Response: In the revised manuscript, we have rephrased the statement as blow:

“Loss of SDHAF4 in the heart led to severe cardiac abnormalities in mice. The pathogenesis involved several common molecular processes which also occur in disease models including DCM, raising the possibility that certain clinical DCM may be associated with mutations or deregulation of SDHAF4”

Line 339/40: I don't agree that the presented study actually infers a clinical need for evaluation of SDHAF4 mutations in patients with DCM. The presented results surely warrant further investigation into the topic, where all SDH related genes should be investigated, not only SDHAF4.

Response: In the revised manuscript, we have rephrased the statement as blow:

“...we predict that mutations of genes encoding complex II components, including SDHAF4, and/or dysregulation of their expression may be associated with cardiomyopathy in human patients.”

1. Ge C, He Y. A Novel Gene Signature to Predict Survival Time and Incident Ventricular Arrhythmias in Patients with Dilated Cardiomyopathy. *Dis Markers* **2020**, 8847635 (2020).
2. McKirnan MD, *et al.* Metabolomic analysis of serum and myocardium in compensated heart failure after myocardial infarction. *Life Sci* **221**, 212-223 (2019).
3. Burke MA, *et al.* Molecular profiling of dilated cardiomyopathy that progresses to heart failure. *JCI Insight* **1**, (2016).
4. Sin J, *et al.* Mitophagy is required for mitochondrial biogenesis and myogenic differentiation of C2C12 myoblasts. *Autophagy* **12**, 369-380 (2016).
5. Crupi AN, *et al.* Oxidative muscles have better mitochondrial homeostasis than glycolytic muscles throughout life and maintain mitochondrial function during aging. *Aging (Albany NY)* **10**, 3327-3352 (2018).
6. Sun N, Malide D, Liu J, Rovira, II, Combs CA, Finkel T. A fluorescence-based imaging method to measure in vitro and in vivo mitophagy using mt-Keima. *Nat Protoc* **12**, 1576-1587 (2017).
7. Holscher M, *et al.* Unfavourable consequences of chronic cardiac HIF-1alpha

stabilization. *Cardiovasc Res* **94**, 77-86 (2012).

8. Semenza GL. Hypoxia-inducible factor 1 and cardiovascular disease. *Annu Rev Physiol* **76**, 39-56 (2014).

REVIEWER COMMENTS

Reviewer #1 (Remarks to the Author):

This is the first time I have reviewed this manuscript. I was requested to determine if the concerns expressed by R1 were satisfactorily addressed.

To confirm validity of the SDHAF4 KO mouse to the clinically relevant condition of myocardial infarction, R1 requested: 1. comparative profiling of metabolic changes after MI and in the KO mice; 2. demonstration that SDH fails to assemble; and 3. evidence that fumarate can mitigate post MI heart failure.

The authors respond with RNA and protein data showing downregulation (not ablation/abrogation) of SDHAF4 expression late after MI. The authors present public dataset data suggesting that this is common finding in human ischemic heart disease and dilated cardiomyopathy. The authors do not really address the important questions of SDH assembly after MI or whether fumarate supplementation improves post MI heart failure. Thus, this manuscript is a description of an interesting but artificial mouse model with new basic science information having uncertain relevance to the human condition.

R1 was concerned about the methodology employed in mitochondrial assays, which the authors have repeated with no change in results or conclusions.

R1 noted that the mitophagy studies were suboptimal in technique, reagents, and the cell lines studied. Authors performed additional studies with mito-keima in H9C2 rat embryonic cardiomyoblast cells, concluding that mitophagy increases as SDHAF4 decreases. Intriguingly, mitochondrial biogenesis and mtDNA levels decreased with SDHAF4 ablation. Thus, the interesting question of how mitochondria number increases in SDHAF4 KO cardiac myocytes seems unanswered.

Minor concerns about histone methylation, heart rate, and English utilization throughout the manuscript seem to have been addressed.

In my opinion the authors have failed to demonstrate that their KO model describes pathophysiology of MI or post MI heart failure. They have also not adequately explained a major aspect of the KO phenotype, increased mitochondria. Most importantly, the link between SDHAF4 expression level, perturbed myocardial metabolism, and cardiac failure is not convincing. Actual side-by-side and well controlled comprehensive metabolomic and transcriptional profiling of post MI and SDHAF4 mice, while correlative, could increase the translational significance of this work.

Reviewer #3 (Remarks to the Author):

The revised version of the manuscript has addressed all the points I have raised previously. I recommend publication in its current form.

Response to reviewer comments

Reviewer #1 (Remarks to the Author):

This is the first time I have reviewed this manuscript. I was requested to determine if the concerns expressed by R1 were satisfactorily addressed.

To confirm validity of the SDHAF4 KO mouse to the clinically relevant condition of myocardial infarction, R1 requested: 1. comparative profiling of metabolic changes after MI and in the KO mice; 2. demonstration that SDH fails to assemble; and 3. evidence that fumarate can mitigate post MI heart failure.

The authors respond with RNA and protein data showing downregulation (not ablation/abrogation) of SDHAF4 expression late after MI. The authors present public dataset data suggesting that this is common finding in human ischemic heart disease and dilated cardiomyopathy. The authors do not really address the important questions of SDH assembly after MI or whether fumarate supplementation improves post MI heart failure. Thus, this manuscript is a description of an interesting but artificial mouse model with new basic science information having uncertain relevance to the human condition.

Response:

Thanks for reviewer's comments. We all understand the complexity of cardiovascular diseases. The rodent models have been widely used in cardiovascular research, whereas the ones, either genetic models or surgery models, which could perfectly mimic the cardiomyopathy conditions in human patients, are very rare. Yet, the studies using mouse models have been advancing our understanding of the pathogenesis, progression, and mechanisms underlying the heart disorders, and offering important insights and clinical implications.

In our present study, a widely accepted MI model was employed to discover the downregulation of SDHAF4 in stressed heart, which was also observed in human patients (Fig. 1). Abrogation of *Sdhaf4* gene in mice resulted in cardiac hypertrophy, enlarged heart chambers, elevated fibrosis and impaired heart function (Fig. 2-3), which mimic the pathophysiologic changes associated with cardiac remodeling and progressive dilated cardiomyopathy in human patients. Though the pathophysiological role of SDHAF4 in mammals is largely insufficient, its molecular function was well established since its discovery in yeast^{1,2, 3}. SDHAF4 was known to bind with flavinated SDHA for the dimerization of SDHA and SDHB, and loss of SDHAF4 could directly result in decreased binding activity between SDHA and SDHB in both yeast and mammals^{1,4}, such effect was also confirmed in cardiomyocytes of *Sdhaf4*-KO mice (Fig. 4), All of these demonstrate that SDH fails to assemble when SDHAF4 is downregulated.

We agree with the reviewer that the effect of fumarate supplementation on MI heart failure was an important issue that need to be addressed, so another set of animal study was thereby performed. The MI mice were supplemented with fumarate before and after the surgery. The heart weight/body weight ratio was dramatically increased after MI and sufficiently decreased by fumarate supplement (Fig. 8a). Intriguingly, echocardiography

study showed that fumarate supplement post MI surgery up to 7 days started to present significant improvement on cardiac function including corrected LV mass, EF%, FS%, LVEDV, and LVESV in MI mice (Fig.8b-e). Further qPCR analysis of hypertrophic and fibrotic associated genes in left ventricles consolidated the benefits of fumarate supplement in maintaining cardiac function after MI surgery. Collectively, the presented observations demonstrate that fumarate can mitigate post MI heart failure and suggest that SDHAF4 deficiency-associated fumarate loss in MI condition plays vital role during the progression of cardiac dysfunction. We have included the new evidences in revised manuscript.

R1 was concerned about the methodology employed in mitochondrial assays, which the authors have repeated with no change in results or conclusions.

Response: We appreciate reviewer's affirmation.

R1 noted that the mitophagy studies were suboptimal in technique, reagents, and the cell lines studied. Authors performed additional studies with mito-keima in H9C2 rat embryonic cardiomyoblast cells, concluding that mitophagy increases as SDHAF4 decreases. Intriguingly, mitochondrial biogenesis and mtDNA levels decreased with SDHAF4 ablation. Thus, the interesting question of how mitochondria number increases in SDHAF4 KO cardiac myocytes seems unanswered.

Response: As we have presented in the revised manuscript, the dramatically increased mitochondrial number in the KO cardiocytes is directly attributed to increased mitochondrial fission driven by p-Drp1^{S616}, which also promotes the progression of mitophagy (Fig. 6). The regulatory role of Drp1 in mitochondrial fission relies on its posttranslational modification status. Phosphorylation at Ser616 is required for its pro-fission activity^{5, 6, 7, 8}, whereas Ser637 phosphorylation inhibits mitochondrial fission⁹. Consistent with these findings, we showed increased p-Drp1^{S616} levels in the hearts of *Sdhaf4*-KO mice at the early postnatal stage along with lower p-Drp1^{S637} levels, supporting the TEM observation of increased mitochondria numbers and qPCR analysis of decreased mtDNA in the knockout mice.

Minor concerns about histone methylation, heart rate, and English utilization throughout the manuscript seem to have been addressed.

Response: We appreciate reviewer's affirmation.

In my opinion the authors have failed to demonstrate that their KO model describes pathophysiology of MI or post MI heart failure. They have also not adequately explained a major aspect of the KO phenotype, increased mitochondria. Most importantly, the link between SDHAF4 expression level, perturbed myocardial metabolism, and cardiac failure is not convincing. Actual side-by-side and well controlled comprehensive metabolomic and transcriptional profiling of post MI and SDHAF4 mice, while correlative, could increase the translational significance of this work.

Response: We appreciate reviewer's comment and suggestion. We compared the differentially expressed gene (DEG) list in the *Sdhaf4*-KO hearts with previously reported gene profiling datasets of MI¹⁰. Notably, the expression of 31.07% (749/2420) of the upregulated genes in MI (MI_3 d) was increased in our mutants (Fig. S4b). Among the

downregulated genes in the MI model, 39.53% (1065/2694) also displayed reduced expression in *Sdhaf4*-KO hearts (Fig. S4b). In particular, downregulation of TCA cycle-related genes was consistently observed in both datasets. We also resurveyed these TCA genes in a profiling data of CM models¹¹. Intriguingly, these genes were consistently downregulated in DCM hearts (Table S1). The results suggest that similar or common molecular processes occur in MI, DCM, and *Sdhaf4*-KO hearts, further indicating that metabolic deregulation in the mutant hearts may be one of the key processes relevant to cardiac pathogenesis. As the direct product of SDH catalysis, loss of fumarate presents as the key feature in the heart of *Sdhaf4*^{Ckmm} mice. Notably, the down-regulation of fumarate and an increase of succinate level were also reported in diseased hearts including MI^{12, 13}, indicating that fumarate associated metabolic deregulation may be one of the key processes relevant to cardiac pathogenesis. Especially, the supplement of fumarate was found to significantly improve cardiac function in MI mice (Fig. 8), further increasing the translational significance of this work.

Yet, we would never take the position to claim that defect of SDH assembly is the only factor that the heart diseases are attributed to. There definitely are multiple factors and a long way to go to elucidate the mechanisms underlying cardiac pathogenesis, what we have been trying to do is to present some new evidence as putting more pieces into the puzzle.

1. Van Vranken JG, *et al.* SDHAF4 promotes mitochondrial succinate dehydrogenase activity and prevents neurodegeneration. *Cell Metab* **20**, 241-252 (2014).
2. Cogliati S, Lorenzi I, Rigoni G, Caicci F, Soriano ME. Regulation of Mitochondrial Electron Transport Chain Assembly. *J Mol Biol* **430**, 4849-4873 (2018).
3. Moosavi B, Berry EA, Zhu XL, Yang WC, Yang GF. The assembly of succinate dehydrogenase: a key enzyme in bioenergetics. *Cell Mol Life Sci* **76**, 4023-4042 (2019).
4. Wang X, *et al.* Hepatic Suppression of Mitochondrial Complex II Assembly Drives Systemic Metabolic Benefits. *Adv Sci (Weinh)*, e2105587 (2022).
5. Jahani-Asl A, *et al.* CDK5 phosphorylates DRP1 and drives mitochondrial defects in NMDA-induced neuronal death. *Hum Mol Genet* **24**, 4573-4583 (2015).
6. Kashatus JA, *et al.* Erk2 phosphorylation of Drp1 promotes mitochondrial fission and MAPK-driven tumor growth. *Mol Cell* **57**, 537-551 (2015).
7. Zhan L, *et al.* Hypoxic preconditioning attenuates necroptotic neuronal death induced by global cerebral ischemia via Drp1-dependent signaling pathway mediated by CaMKIIalpha inactivation in adult rats. *FASEB J* **33**, 1313-1329 (2019).
8. Xu S, *et al.* CaMKII induces permeability transition through Drp1 phosphorylation during chronic beta-AR stimulation. *Nat Commun* **7**, 13189 (2016).

9. Yu R, *et al.* The phosphorylation status of Ser-637 in dynamin-related protein 1 (Drp1) does not determine Drp1 recruitment to mitochondria. *J Biol Chem* **294**, 17262-17277 (2019).
10. Zhang F, *et al.* Long noncoding RNA Cfast regulates cardiac fibrosis. *Mol Ther Nucleic Acids* **23**, 377-392 (2021).
11. Burke MA, *et al.* Molecular profiling of dilated cardiomyopathy that progresses to heart failure. *JCI Insight* **1**, (2016).
12. Wang X, *et al.* Metabolic Characterization of Myocardial Infarction Using GC-MS-Based Tissue Metabolomics. *Int Heart J* **58**, 441-446 (2017).
13. Chouchani ET, *et al.* Ischaemic accumulation of succinate controls reperfusion injury through mitochondrial ROS. *Nature* **515**, 431-435 (2014).

REVIEWERS' COMMENTS

Reviewer #1 (Remarks to the Author):

The manuscript is improved, especially by the fumarate studies. I believe that the expressed concerns from original R1 have been adequately addressed.

Response to reviewer comments

Reviewer #1 (Remarks to the Author):

The manuscript is improved, especially by the fumarate studies. I believe that the expressed concerns from original R1 have been adequately addressed.

Response: We appreciate reviewer's affirmation.